

# Design and implementation of an intelligent sports management system (ISMS) using wireless sensor networks

ZhiGuo Zhu

College of Physical Education, Luoyang Normal University, Henan, China

## ABSTRACT

In recent years, growth in technology has significantly impacted various industries, including sports, health, e-commerce, and agriculture. Among these industries, the sports sector is experiencing significant transformation, which needs support in accurately monitoring athlete predicting and performance injuries arising due to traditional methods' limitations. Keeping the above in mind, in this article, we present the Intelligent Sports Management System (ISMS) with the integration of wireless sensor networks (WSNs) and neural networks (NNs), which enhance athlete monitoring and injury prediction. Our proposed ISMS consists of several layers: user interface, business logic layer, data management layer, integration layer, analytics and AI layer, IoT layer, and security layer. To facilitate interactions for athletes, coaches, and administrators, our planned ISMS integrates a user-friendly interface accessible through web and mobile applications. Besides, scheduling and event management are managed by the business logic layer. Similarly, the data management layer can process and store comprehensive data from various sources. To ensure smooth data exchange, the integration layer connects the ISMS with third-party services, and the analytics and AI layer leverages machine learning to provide actionable insights on performance and outcomes. In addition, the IoT layer collects real-time data from sensors and wearable devices, which is essential for performance analysis and injury prevention. Finally, the security layer ensures data integrity and confidentiality with robust encryption and access controls. To evaluate the system performance in different scenarios, we performed many experiments, which show that the proposed ISMS model shows the system efficacy in improving accuracy (0.94), specificity (0.97), recall (0.91), precision (0.93), F1 score (0.95), mean absolute error (MAE) (0.6), mean square error (MSE) (0.8), and root mean square error (RMSE) (0.9), compared to traditional methods. From these results, it is clear that our suggested approach improves athlete performance monitoring, injury prevention plans, and training schedules by presenting a complete and novel solution for recent sports management.

# INTRODUCTION

These days, information and communication technology plays a key role in every field of life. The rapid development of this technology has significantly affected various fields,

Corresponding author
ZhiGuo Zhu,
zhuzhiguo@lynu.edu.cn

including sports, through the integration of wireless sensor networks (WSNs). Traditionally, collecting physiological and biomechanical data, such as heart rate, body temperature, and muscle activity, has involved manually attaching sensors to athletes at specific intervals (*Yang & Lv, 2020*). However, WSNs have changed this process by enabling continuous, real-time monitoring. This change allows for more accurate assessments of athletes' performance and health by facilitating timely adjustments to training programs and assessments (*Bashkaran et al., 2023*). Through early detection of potential problems, WSNs help prevent acute injuries and manage joint or muscle strain, extending athletes' careers and enhancing their overall health (*Si & Thelkar, 2024*). Moreover, WSNs support rehabilitation by enabling ongoing monitoring and adjustment of exercise regimens, which helps speed recovery and ensures safety without direct supervision (*Li, Kumar & Alazab, 2022*). As WSN technology evolves, more innovations are expected to enhance athletic performance and health management.

In addition, WSNs have become a notable improvement in sports management and align with the progressive data revolution and personalized athlete attention. The feasibility of getting large datasets in real-time changes the training strategies and opens the possibility of making individual adjustments and preventing injuries (*Zhang, Chai & Li, 2024*). In real-time, WSNs enable the observation of the athlete's physiological and biomechanical data to observe a signal of an upcoming physical problem and quickly fix it so that athletes stay at their best all the time. Moreover, technology in sports improves the training process since information on muscle activity and weariness is readily available, thus improving injury preventive measures and total performance (*Qi et al., 2024*). Applying WSNs in sports management is beneficial in enhancing health indicators through more precise and timely capturing of health parameters by designing effective training processes that might meet the special individual requirements of the athletes. This technology is developing daily, and it opens new possibilities for applying it in sports management, which may provide more extensive and more effective methods of performance and health controls (*Edriss et al., 2024*). When integrated with machine learning (ML), convolutional neural networks (CNNs) and deep neural networks (DNNs), WSNs help perform complex analyses of the collected data, which highlights vital performance metrics and possible issues.

Besides, WSN and machine learning algorithms have become widely used to analyze the data collected by such a network. These algorithms enhance decision-making concerning injury risks, recovery treatments, and training schedules (*Yang et al., 2024*). It is recognized that machine learning offers advantages over conventional techniques and strategies since it enhances athletes' results and health. Such algorithms apply to the audio, video, and textual data from matches and provide a deeper understanding of team player behaviors, tendencies, and strategic actions by enhancing the spectator's experience (*Herberger & Litke, 2021*). WSNs increase the efficiency of athlete management, decrease the occurrence of injuries, and boost fan relations by offering more comprehensive and precise data with the help of a combination of machine learning (*Abramov et al., 2021*). In this generation, the impact of such technologies is still unexplored. The implications of their use are expected to be more profound in athletes' performance and health management (*Al-Asadi*

*& Tasdemır, 2022*). Current sports management systems face some problems, like inaccuracy of the data, the lack of real-time analysis, and problems with integration. The conventional methods are primarily based on offline methods of data collecting, and these are often invasive and narrow in scope. Such systems may encounter difficulties giving output in real-time and effectively responding to changes in the conditions as they are based on time-series data collection and essential analytical tools.

Based on the above, this article presents the development of an intelligent sports management system (ISMS) that employs WSNs and neural networks for design and implementation. This framework aims to improve the precision, robustness, and speed of athlete performance monitoring, rehabilitation, and sports management services by integrating WSNs and neural networks. In addition, this study is intended to contribute significantly to the constant development of international management of sports initiatives. The combination of WSNs, the continuous monitoring of athletes remotely, and the coaches and medical personnel are arranged to have important physiological and biomechanical parameters at their disposal. The latter is enhanced by neural networks, which help provide meaning patterns and trends for strategic decision-making in sports management from large datasets. The other main goals of the article are the following:

- Firstly, this article identifies an innovative, ISMS combining WSNs and neural network algorithms. This approach allows observing athletes' physiological and biomechanical parameters non-stop and in real-time, which was impossible with the help of the conventional approach of data gathering.
- Second, it establishes an extensible system design that adopts low-power wireless communication like Zigbee and Bluetooth Low Energy for proper and effective data transfer of wearable sensor nodes. This integration minimizes interruption and events and enhances the standard of the obtained data, which was a disadvantage because of some periodic and disturbing data collection techniques.
- Thirdly, it uses the machine learning algorithms in CNNs and DNNs to analyze the obtained data. These techniques increase the ability of the system to analyze the patterns at a finer level and identify emerging trends, performance prognosis, and possible risks for injuries with recommendations and solutions unique to athletes.
- Finally, it builds robust evaluation criteria to measure the performance of the ISMS with particular attention to the system's efficiency, the correctness of data, the satisfaction level of users, and real-time monitoring. This approach guarantees the system's effectiveness in accomplishing the population's needs, which presents flexibility in sports management.

The rest of the article is organized in logical order as follows: the Literature Review section of the article represents the literature work with multiple recent studies; System Design and Architecture of the Intelligent Sports Management System section presents the proposed methodology of the system design and architecture of the ISMS; the Application of Sensor Networks and Neural Networks in Enhancing Athlete Performance section

describes the sensor networks and neural networks implementation to centralized databases and cloud storage, leading to performance analysis; the Experimental Results and Analysis section of the article represents of the experimental results and analysis of the implementation of an ISMS. Finally, the conclusion is described in the Conclusions section of the article, which includes future work and direction.

## LITERATURE REVIEW

These days, developing WSNs and neural networks have impacted sports management, especially monitoring athletes and assessing their performances. WSNs are incorporated into ISMS as a new technological improvement in sports technology, as its primary goal is to increase the achievement of athletes and the efficiency of management. WSNs help in the real-time acquisition of various forms of data from the wearable sensors incorporated in athletic wear concerning biomechanics, physiological conditions, and performance. This section discusses the existing literature and previous studies concerning the design and deployment of ISMS, emphasizing WSNs.

Technological improvements have significantly enhanced the ability to monitor and analyze athletes' performance in real-time. Combining wearable technologies and neural network algorithms has become pivotal in advancing sports management and performance evaluation. *Zheng et al. (2024)* analyzed wearable technologies with a particular focus on their application in professional sports contexts as an application for real-time athlete monitoring. It highlighted that WSNs are instrumental in collecting accurate physiological and biomechanical data. Improving the training schedules to the ideal and the performance gains in sports is useful. The study also points to the extraordinary capacity of WSNs to enhance the monitoring of athletes and assume a critical role in the evolution of sports management. *Ding et al. (2023)* highlighted the interaction between IoT devices and AI methods enriched with neural network algorithms by improving sports performance observation. This work refers to the use of WSNs for steady monitoring of data and AI for better processing of the data to enhance the decision-making system in sports. According to this research work, the technologies are intended to be used to develop performance monitoring strategies and discover how analytics will likely transform the processes of athlete management and training methods. *Gu et al. (2024)* analyzed some of the numerous applications of WSNs in sports, mainly focusing on monitoring athletes' performance and the environment. According to their work, WSNs improve decision-making and management effectiveness through real-time data acquisition and monitoring, increase training efficiency, and prevent injuries in athletic fields. They have compared different neural network models for predicting an athlete's performance based on data gathered from WSNs. This work quantitatively examines these models' validity and efficiency in predicting performance aspects like speed, strength, and power. According to *Xu et al. (2021)*, training different Neural Network architectures and their related areas will allow for better adaptation of training regimens to improve the outcomes of injury prevention strategies. Their implications suggest how coaches and sports scientists can use it to enhance athlete performance and safety.

In addition to the contributions of the above scholars, *Xiao et al. (2017)* claimed that IoT gadgets enable differentiation of the likelihood of incurring an injury and play a crucial role in extending athletes' performance tenures. According to them, timely corrective actions and adjustments in training schedules are due to the detection of critical physiologic and biomechanical markers, which reduce the chances of body injuries and promote athletes' future health and performance. Similarly, *Uddin et al. (2024)* analyzed machine learning algorithms usage in real-time sports analytics based on the data gathered from WSNs work to draw useful information to improve game strategies and review players' performance. This research work enquired about the progress of using machine learning in decision support systems, where sports coaches and analysts can utilize complex methods to make tactical choices and increase team effectiveness and individual athletes' evaluation. They also highlighted the enhancement of M2M learning with WSN in sports management. The training methods of the research article offer (*Tian et al., 2023*) timely information and data. In addition, timely prevention of injuries and treatment measures and optimization of the expected performance of the athletes. The possibilities of wearable technologies, particularly in sports development, point out the changes that will transpire in the training, supervision, and even administration of athletes, which, in one way or another, will improve the efficiency and management of sports. To analyze the literacy of neural networks in sports biomechanics, specifically from WSNs, the research presented by *Zhang et al. (2023)* points out the possibility of using neural network features to recognize complex movement patterns and physiological signals to improve athletic performances, including enhancing the model's performance and incorporating feedback mechanisms. All these advancements seek to deliver real-time results that are useful to athletes and coaches and also aid in improving the training methods, thus thriving a revolution of neural networks in the biomechanics in sports.

The existing and appearing improvements and research on WSNs in athlete health monitoring align with the sensors' evolution and analysis methods to improve the accuracy and scalability of sporting activities. According to *Li et al. (2024)*, the developing WSN technology can offer increased accuracy of the sensors used and the possibility of using techniques of data analysis that will allow the health monitoring of athletes to be achieved with improved accuracy in the proper making of a decision concerning training sessions, performance improvement, and preventing possible injuries. Their findings of the present investigation also speak volumes for the increasing role of WSNs in developing the field of sports medicine in the future. The severe management of sports is done through AI systems, mainly WSNs and NNs. This sub-topic covers advanced technology and its relation to sports performance with precision and accuracy in data tracking and analysis, modeling and understanding, and finally, the training process and personal coaching. The confidentiality of data, especially that of the athletes, and the scale of the problem brings out the need for solutions that are easy to implement and can handle large amounts of data. The analysis of these factors proves that integrating AI and WSN may significantly transform the processes of sports management and performance enhancement, as highlighted by *Dang et al. (2023)*. Similarly, *Yang & Tang (2022)* are carried out through WSN in the framework of IoT and big data analytics for sports performance monitoring.

They critically discuss the practical implementation of real-time analytics in training and injury prevention to collect and analyze enormous data for effective training solutions and improved athletes' performance. The data obtained from IoT interfaces and its correlation with big data analytics offer a significant understanding of an athlete's physical well-being and capability, helping trainers and sports physiologists make evidence-based decisions. According to the research presented by *Tong, Li & Wang (2023)*, Deep learning recognizes athlete activities with data collected from WSNs and other sensors. Their proposed model measures the efficiency of these techniques in outlining and analyzing the athletes' actions and provides a great understanding of their performance. In addition, this model improves the efficiency of training and evaluation of the performance of different deep learning models to enable trainers and coaches to make proper evaluations and regimens that would help improve the performance of athletes. They established the importance of deep learning in dealing with changes in activity recognition and performance analysis with sports.

In light of the achievements of the above researchers, the proposed ISMS goes further by bolstering them with low-energy sensors and utilizing a holistic approach that combines numerous machine-learning algorithms. The system involves energy-saving measures like Zigbee and Bluetooth low energy (BLE), which makes it less demanding and has little data interference. The proposed components are the secure central data core and cloud solutions for further expansion. It suggests that these models improve the accuracy of performance assessment and prognosis for possible injuries using techniques like random forests, SVMs, isolation forests, autoencoders, and neural networks. The current methods, as described in this article, could be more efficient and consistent. The proposed system will deliver better athlete monitoring and management approaches as it is more integrated and scalable. The current technologies provide superior functions for increasing sports performance and avoiding injuries.

## SYSTEM DESIGN AND ARCHITECTURE OF THE INTELLIGENT SPORTS MANAGEMENT SYSTEM

The intelligent sports management system employs emerging technologies to develop a sustainable, centralized, and effective mechanism to facilitate the management of sporting events. ISMS's architecture combines several elements to encourage the servicing of athletes, coaches, administrators, and fans and produce meaningful data to study their practices. Critical architectural layers cover the front-end user interface for web and mobile clients, the business logic layer containing crucial functionalities and APIs, and the data management layer for structured and unstructured data. The integration layer, a component required for interfacing with third parties, is unique, while the analytics and AI layer involves interpreting data or machine learning algorithms. The IoT layer includes real-time information through sensors and wearables; this layer in the framework is the security layer, which protects data and user access. This conceptual design promotes higher intelligence in sports management to boost the field.

## The architecture of the proposed intelligent sports management framework

An ISMS uses WSNs and neural networks to improve athlete tracking. Wearable sensor systems and networks will be composed of sensor nodes that the athletes will wear to record physiological and biomechanical parameters like heart rate, body temperature, acceleration, geographical location, *etc.* These sensors will use low-energy wireless protocols such as Zigbee and Bluetooth Low Energy to reduce the interference of the network while taking long durations before they are required to be charged, which is ideal for monitoring during sports activities. The data is aggregated from the sensor nodes and transmitted safely to other layers, particularly the cloud (*Li et al., 2022*). This hub also ensures security when transferring data without outsiders accessing it. Data storage and analysis will be done on the cloud server, using solid relational databases for data storage and querying. The collected data was analyzed using machine learning approaches for pattern recognition of the data collected, data anomaly detection, and to obtain performance measures. Classification models such as random forest, support vector machine, isolation forest, and autoencoder will determine the performance, exceptional grades, injuries, and trends. Linear regression will depict relations, and neural networks will capture trends and non-linear data features. The state-of-the-art data analysis methods will be improved with the help of CNNs and DNNs, which are typically implemented for image and video data and process motion patterns of visuals (*Khan et al., 2024*). DNNs process big, multifaceted data to find subtle patterns and trends in athletes' performance and well-being. Real-time athlete data captured will help the coaches and medical staff include information on possible injuries or drops in performance. Real-time feedback and threshold notification will also improve the monitoring process, while the training data and recovery point data will significantly benefit the training schedules' outcomes. The assessment of the ISMS will require an ongoing monitoring process to review its effectiveness, including system efficiency, speed of response, throughput, and user satisfaction. Constant updating and fixing align with the feedback and performance data gathered to enhance the system's efficiency and reliability in managing and monitoring the athletes' performance. The enhancements using input and performance data will guarantee that the system continues to operate and is responsible for promoting the athletes' performance, as depicted in Fig. 1.

The data management layer is meticulously designed to uphold data integrity through validation rules, checksums, and transactional consistency. Robust data validation mechanisms are implemented at the input stage to ensure that only accurate and valid data is processed. At the same time, Atomicity, Consistency, Isolation, Durability (ACID) properties are leveraged in database transactions to maintain integrity throughout data operations. The system utilizes a scalable and flexible storage solution capable of efficiently handling large volumes of real-time data generated by multiple athletes. The architecture encompasses structured and unstructured data storage options, employing relational databases such as PostgreSQL or MySQL for structured data. In contrast, unstructured data may be managed using NoSQL solutions like MongoDB or through a data lake

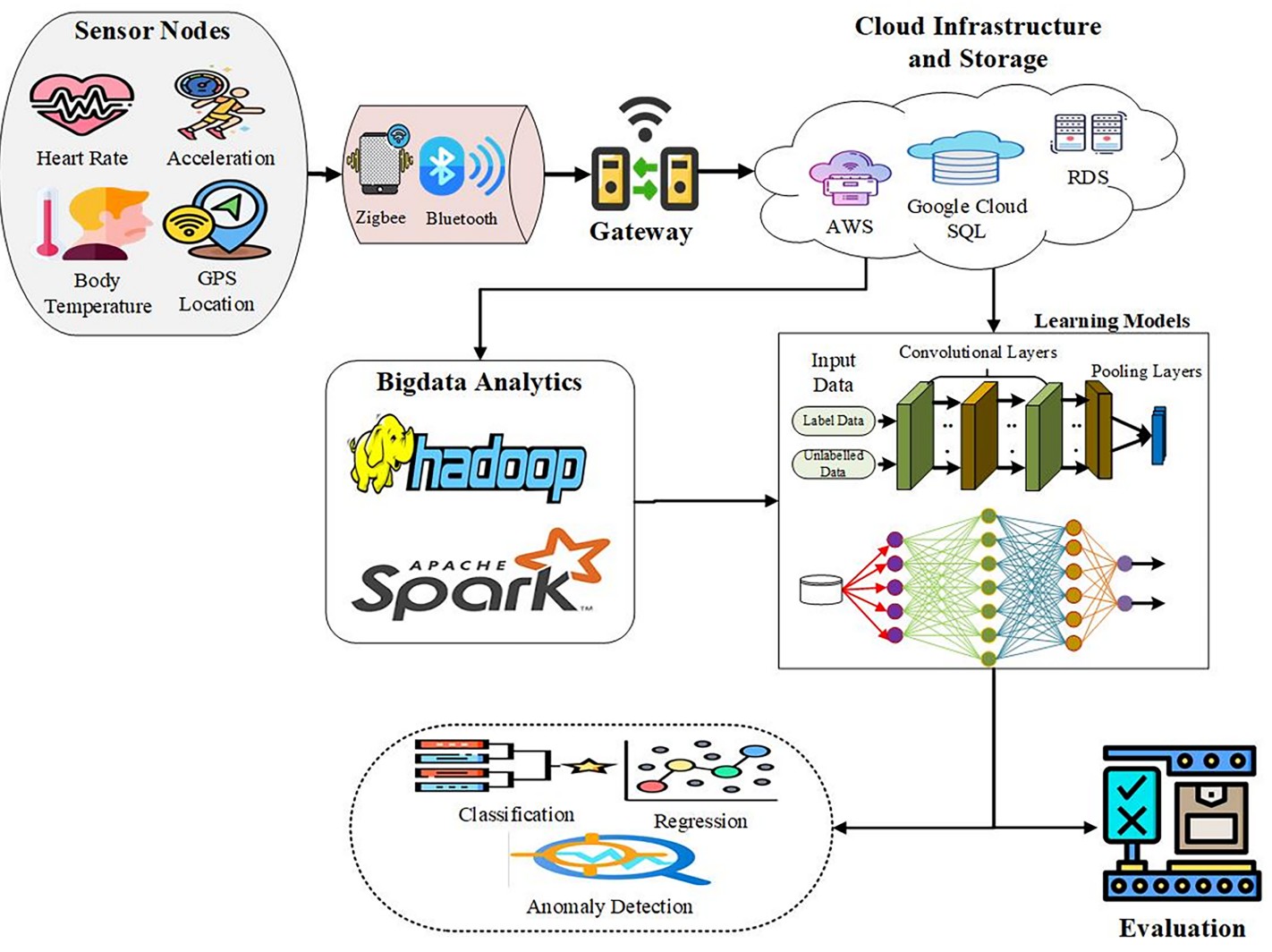

**Figure 1** Architecture of the ISMS integrating WSNs and machine learning.

architecture. To effectively manage the influx of real-time data from various sources, including wearable devices and IoT sensors, a data pipeline architecture is implemented, utilizing stream processing frameworks such as Apache Kafka or Apache Flink to ensure low latency and high throughput for timely analytics and insights. A comprehensive overview of database management techniques focuses on data partitioning and sharding to enhance performance and scalability, efficient indexing strategies to optimize query performance, and robust backup and recovery mechanisms to safeguard against data loss. Additionally, visual diagrams illustrating the data pipeline architecture and data flow through the system clarify how data is ingested, processed, stored, and accessed in real-time.

## Benchmark dataset

To develop a robust and efficient prediction model, a reliable and accurate benchmark dataset is essential, as demonstrated in the recent work by *Burns et al. (2022)*. This benchmark dataset includes three types of data: physiological measurements, self-reported assessments, and behavioral data. The dataset is organized with participants' demographic details, the duration of positive affective behavior coded from video recordings (in seconds), and their responses to the STAI and SPIN assessments. The assessments are provided as a Supplemental File. Additionally, various physiological measurements were collected, including ECG, cardiological impedance, and EDA, from which heart rate (HR), heart rate variability (HRV), and EDA metrics were derived. The data files are structured with individual folders for each participant, containing their pre-processed HRV and EDA data, expressed in Eq. (1).

$$D = \{(X_i, Y_i)\}_{i=1}^{n} \tag{1}$$

where $X_i$ is the feature vector for participant $i$, $Y_i$ is the corresponding label or response, and n is the total number of participants.

## Design of an intelligent athlete motion system based on wireless network

A widely used structure in sensor networks is implemented in systems designed to manage and monitor sports and athletic performance. The architecture includes several essential modules that enable collecting, transmitting, and analyzing relevant data. Key among these is the user terminal, which provides a crucial interface for athletes and coaches. This terminal allows users to view current statistics and receive alerts on performance or conditions, ensuring straightforward and convenient access to data. The terminal's design prioritizes ease of use, allowing clients direct access to the information collected by the sensor network (*Liu & Zhang, 2021*). The network consists of multiple nodes that gather data, with cluster heads overseeing the operations and managing information within their assigned clusters. The data from several IoT and wireless nodes sum up the data and transmit it to the central base station. The IoT Nodes and wireless nodes are spread across the monitoring area, and they are provided to measure necessary parameters, such as the heart rate, body temperature, and movement acceleration. These wireless nodes utilize Zigbee or Bluetooth Low Energy (BLE) standards for data exchange for reliable and low power consumption. The system's components collectively enable essential functionalities, as depicted in Fig. 2. By integrating these elements, the system improves athlete performance and well-being and contributes significantly to sports science, offering advanced tools to optimize athletes' training and recovery.

The base station, the aggregator, manages the interface between the sensor network and the external data structure. It collects data from the cluster heads and sends it to the data hub for processing. The data hub is the system's key component focused on collecting, processing, and storing the gathered data. It employs a combination of cloud servers and databases for large volumes of data to ensure the information is accessed for real-time purposes or historical analysis. Data Transmission across the network implies transferring

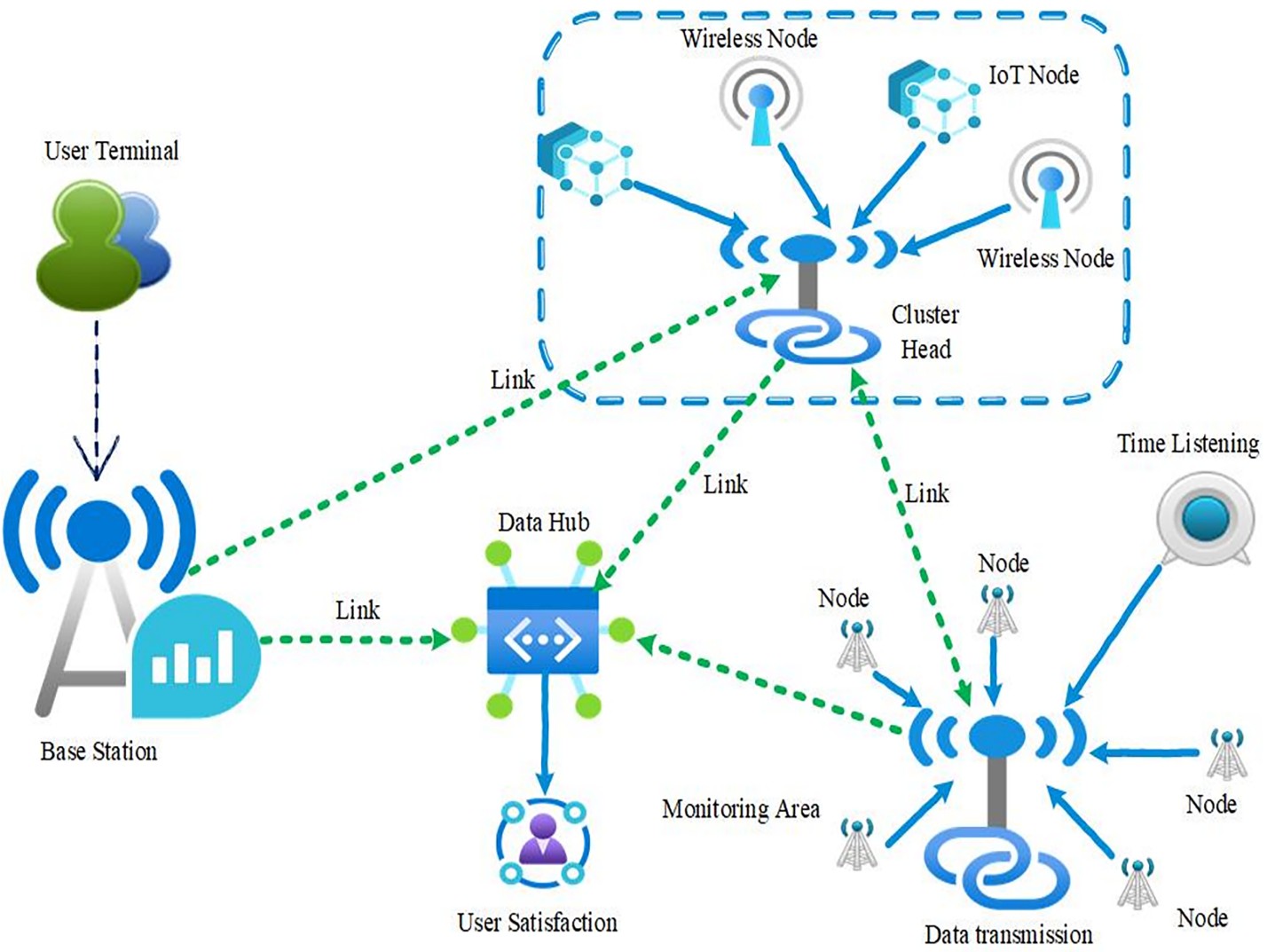

**Figure 2  Architecture of the intelligent athlete motion system based on WSNs.**

data gathered from the sensor nodes from the cluster heads right through to the base station. This activity is done with the help of Time Listening techniques that help manage the signals for data and time collection to ensure the accuracy of the data and avoid its loss. User satisfaction is a metric of the ability of the system being designed. This is more or less an evaluation of how the system supports the needs of its users through features like the accuracy of data, transmission, and the layout of its interface. Mathematical rigor to the intelligent athlete motion system to delve into data transmission efficiency, sensor data processing, and user interface evaluation. Here are some mathematical aspects:

The data is transmitted from the IoT node to the central system. The higher the transmission rate, the faster the data is sent, crucial for real-time monitoring and analysis in sports management systems, as expressed in Eq. (2).

$$R_i = \frac{D_i}{T_i} \; (bit\,per-second,\,BPS) \tag{2}$$

This formula calculates $R_i$ by dividing the total data $D_i$ by the time $T_i$. This yields the transmission rate in bits per second (bps), indicating how fast data is transmitted from the $i^{th}$ IoT node.

To assess the energy efficiency $E_i$ of the $i^{th}$ IoT node during data transmission, use the following Formula (3):

$$E_i = \frac{D_i}{P_i.T_i} (bits\,per\,joule,\,bpJ). \tag{3}$$

In this formula, $P_i \cdot T_i$ represents the total energy consumed, as energy is calculated by multiplying power (in watts) by time (in seconds). Thus, $E_i$ calculates the number of bits transmitted per joule of energy consumed, measuring how efficiently the node uses energy to transmit data.

In a cluster where each cluster head manages $n$ IoT nodes, the total data aggregated by the $i^{th}$ cluster head is given in Eq. (4):

$$D_{total,i} = \sum_{j=1}^{n} D_{ij}. \tag{4}$$

In this equation, $D_{(total,\,i)}$ is computed by summing the data from all $n$ IoT nodes within the $i^{th}$ cluster, reflecting the total amount of data that the cluster head needs to process and forward.

The moving average filter calculates the average of the data points within a specific window size $w$ centered around the current time $t$. It helps in smoothing out the time series data by averaging the values over a defined window, which reduces noise and provides a clearer view of trends. To compute the moving average $MA(t)$ at time t for a time series data $x(t)$ with a window size w, use the following are expressed in Eq. (5):

$$MA(t) = \frac{1}{w} \sum_{k=t-w+1}^{t} x(k). \tag{5}$$

In the above formula, $MA(t)$ of the moving average is the average of the activities in the time frame of w just up to time t, thus helping to make the chart bear and easy to analyze the characteristics patterns of the respective time series.

User satisfaction reflects how well the system meets user expectations based on three key aspects: the vagueness of the data entered, the dependability of data communications, and the friendliness of the system's interface. Each element above is given a specific value indicating its significance to the overall satisfaction. The weighted factors are then summed up to get the final user satisfaction score. To model user satisfaction S, use the following are expressed in Eq. (6):

$$S = w_\alpha A + w_\beta R + w_\lambda U. \tag{6}$$

In this equation, *S* is calculated by summing the weighted contributions of data accuracy *A*, transmission reliability *R*, and interface usability *U*. Each factor's contribution to the overall satisfaction score is determined by its respective weight, which reflects its relative importance.

The total data aggregated by a cluster head is the sum of the data collected from all the IoT nodes managed by that cluster head. Each cluster head collects data from multiple IoT nodes in a sensor network. To find the total amount of data aggregated by the cluster head, you sum the data collected from each node within its cluster. This aggregated data represents the total volume of information the cluster head needs to process and transmit further, expressed in Eq. (7).

$$D_{total} = \sum_{j=1}^{n} D_j = \sum_{j=1}^{\eta} \delta_j. \tag{7}$$

In both representations, $D_{(total)}$ is computed by summing the data from all the nodes under the cluster head, reflecting the cumulative data that needs to be managed by the cluster head.

## Integrated sports management system with advanced data security

The structure used in an ISMS using wireless sensor networks is multi-layered to address the issues of data handling, processing, and security. The data management layer's role is to gather and preserve information and perform initial data processing. This layer accumulates data received from wireless sensors in real-time and includes physical characteristics such as heart rate and acceleration. Pre-cleaned and well-organized data is ready for quick analysis with possibilities for future analysis; cloud storage is available if needed. For data transmission security, transport layer security (TLS) is used to encrypt the communication between sensors and central servers to make the athlete's data untraceable by a third party. In support of this, the security layer adds to data security by using encryption techniques, authentication, and access control only for approved users. It also features detailed levels of protection, such as an element of being able to detect an anomaly that warns against unlawful access. In aggregate, this layered approach helps maintain that ISMS remains a secure and orderly system efficiently used in sports management.

## Wireless sensor network-based ISMS

It developed networks of spatially dispersed, self-organized sensors to gather data from the environment and share it wirelessly. In a sports management system, WSNs track parameters of athletes' activity levels, heart rate, and other physiological data. Integrating and managing intelligent sports using WSNs involves different parts to help improve athletes' performance and health analysis. The so-called sensor nodes—small, self-contained devices containing sensors—are installed on the athletes' bodies to monitor some of the given parameters, including, for example, speed, pulse rate, and other physical characteristics. These sensor nodes, in turn, transmit the acquired data to a base station by wireless means and then to a data management center (*Guembe et al., 2021*). Collecting

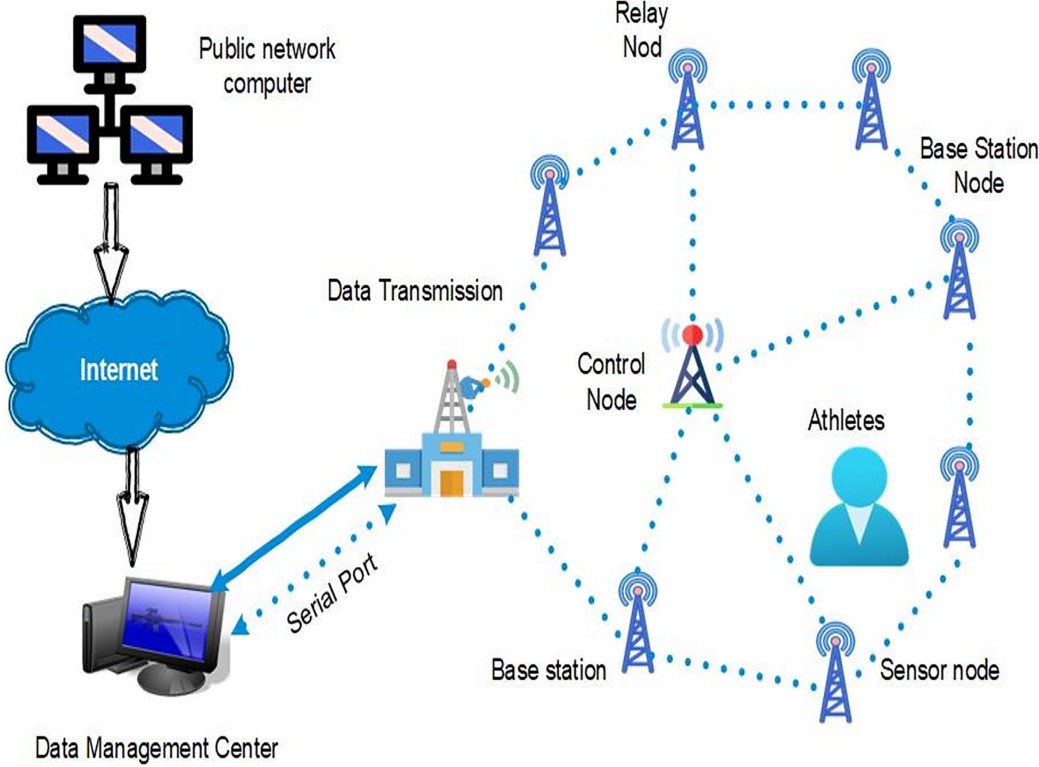

**Figure 3 System design for wireless sensor network-based athlete monitoring and data management.**

data requires an efficient data management center that processes and analyzes the data with the help of sophisticated tools and offers its interpretations. The information is attained by users such as the coaches and athletes *via* a public network computer that connects to the internet, whereby the base station, the data management center, and the remote users are linked. Moreover, serial ports can be used for the first data acquisition or low data rate operations, and data transmission standards guarantee that data is transmitted through the network as effectively and faultlessly as possible. The integrated system enables exhaustive performance monitoring, health condition assessment, and informed decision-making within sports management and athlete training. This structure empowers coaches and trainers to track key performance metrics, assess athletes' physical conditions, and make data-driven decisions to enhance training effectiveness and overall athletic outcomes, as illustrated in Fig. 3.

## Application of sensor networks and neural networks in enhancing athlete performance

Applying sensors and neural networks to improve athlete performance marks a transformative advance in sports technology. WSN produces real-time, detailed data on physiological and biomechanical metrics such as heart rate and muscle activity, which are crucial for monitoring and optimizing athlete performance. With their complex data analysis and pattern recognition capacity, neural networks process this data to provide

actionable insights. This integration enables personalized training programs, proactive injury prevention, and improved performance. These technologies lead to innovative strategies for enhancing athlete performance and well-being through advanced data analysis and interpretation.

## Sensor networks leading to performance analysis

The principles of WSN and the ability to implement an ISMS entail combining elements to promote athletic capability and the athlete's health. At the center of this system is the player, sensors that record the player's movements and physiological data such as heart rate. These sensors are worn on the athletes to be in a position to capture data every time the athlete is participating in an activity. The gathered data is sent through wireless connections, which store all the incoming information and data received from the centralized database, which is then smoothly shifted to cloud storage (*Zhen, Kumar & Samuel, 2021*). Cloud storage allows for secure, expandable resources and easy access to big data, thus making it easy for big data analysis. The players' data are collected and sent to the centralized database and cloud storage and then to the performance analysis, injury prediction, and training optimization modules as shown in Fig. 4. Incorporating these elements guarantees a coordinated and efficient functioning of management in sports that impacts both the physical exercise of athletes and the administration of their health.

The fundamental activities of the system are related to performance analysis, injury prediction, and training optimization. The examination helps analyze the compiled statistics on athletes' achievements and possible drawbacks and explain their effectiveness. To predict the chances of an injury and help prevent one by employing data from past incidences and constantly updated data. Training optimization involves developing specific training strategies for a particular training need in an organization through data that would further improve training efficiency. These interfaces are integrated into the smartphone systems so players and coaches can easily view information and trends. With the help of a phone, athletes can see the player profiles containing the necessary data on performance and health status (*Miah et al., 2022*). The profiles, such as the team profile, give an overview of the players' performance and enhance the capability of coaches to track the team's activities. Coaches can use real-time alerts to check on any alarming circumstances the system identifies.

In ISMS, WSNs play a crucial role by collecting and processing data to enhance athletic performance and health monitoring. Adjacent nodes within these WSNs can often identify similarities in the data they gather, leading to significant data redundancy. This redundancy, occurring in a specified random area $S$ with a distribution density $P(a, b)$, is critical in ISMS. The redundancy level between the nodes is given in Eq. (8):

$$\eta = \xi \cdot SeP \tag{8}$$

where η represents the degree of redundancy, ξ is a coefficient related to redundancy, and *SeP* denotes the spatial density of the data points. If the highly redundant raw data are transmitted directly through *SeP*, it could lead to excessive energy consumption by the nodes, which is a significant concern in the ISMS.

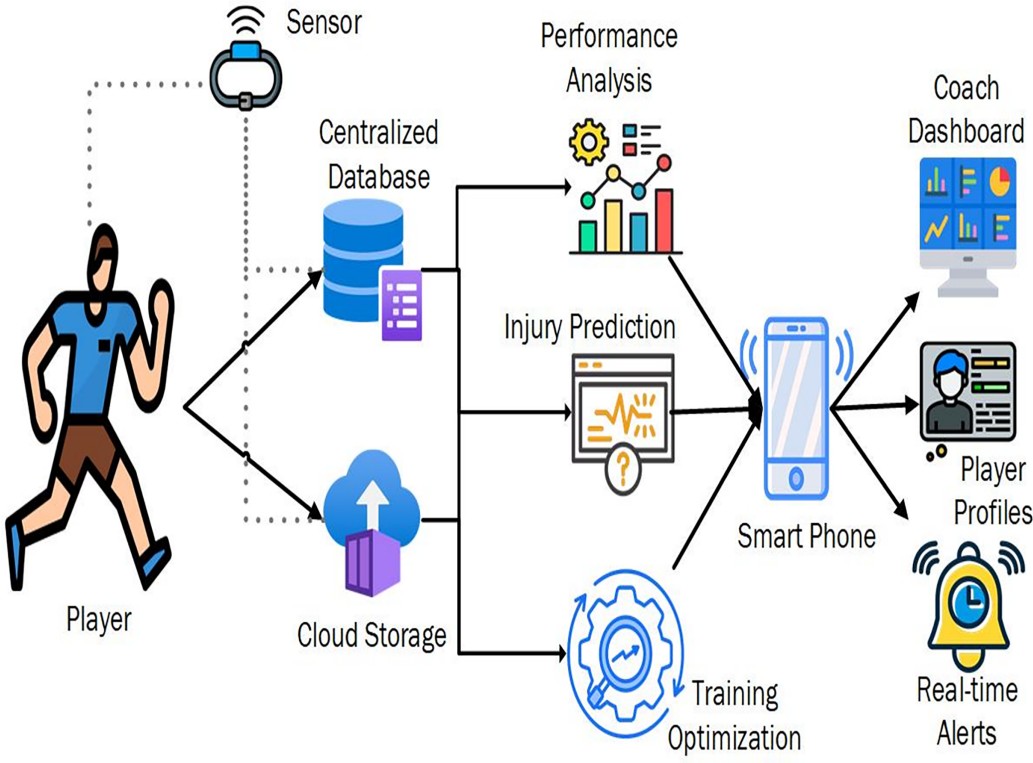

**Figure 4 System framework for coaches and players in an ISMS.**

To address this issue, beamforming algorithms are utilized within the ISMS. These algorithms are designed to filter and combine signals from multiple sensors efficiently. The general form of the beamforming algorithm used in such systems is expressed in Eq. (9):

$$b = \sum_{i=1}^{n} \sum_{l=1}^{n} w_i(l) S_i(n-1). \tag{9}$$

Here, $S_i(n)$ represents the signal collected by the ith sensor, and $w_i(l)$ denotes the weight filter for the ith sensor. Various beamforming methods, including the minimum mean square error (MMSE) method and the maximum energy beamforming algorithm, are employed to optimize filter performance in the ISMS. These methods balance node energy consumption and overall performance, reducing energy use while enhancing data accuracy and reliability.

Accurate time synchronization is vital for the efficient operation of ISMS. The difference between the frequency $f(t)$ of a sensor node's clock and the nominal value of 1, known as clock drift, is expressed as Eq. (10):

$$p(t) = f(t) - 1. \tag{10}$$

The local clock reading $C_i(t)$ for a sensor node in the ISMS at time $t$ is given by Eq. (11):

$$C_i(t) = \frac{1}{f_0}\left(\int_{t_0}^{t} f(t)dt\right) + C_i(t_0). \tag{11}$$

where $t_0$ is the starting time, and $C_i(t_0)$ is the clock reading at $t_0$. Due to potential discrepancies from manufacturing errors in crystal oscillators, $f_0$ and $f(t)$ may not always be equal. Therefore, the local clock in the ISMS can also be represented as Eq. (12):

$$C_i(t) = x_i(t - t_0) + b_i \tag{12}$$

where $b_i = C_i(t_0)$ is the initial clock reading, this formulation helps manage variations and ensures precise time synchronization across the ISMS.

## Architecture and functionality of neural networks

A neural network is a complex and protracted mathematical model that tries to mimic the human brain's operation to identify the relationships between cause and effect and make correct predictions for any data set. The neural network structure forms layers that consist of several neurons that connect. The initial data and features are introduced into the input layer, and the hidden layers analyze this information using connection weights and bias. Every neuron in these layers calculates a weighted sum of its inputs, adds a bias, and applies an activation function, which is an added non-linearity. Softmax works for binary classification and outputs values in the range of 0 and 1; ReLU, which outputs values starting from 0 and going up towards infinity, helps build layers, while tanh helps get smooth gradients ranging between −1 and 1.

When forward propagating data in a given network, data moves from the input layer to the hidden layers and subsequently to the output layer, where predictions or classifications are made. The performance of such predictions is evaluated from the loss function such as MSE for regression problems or cross-entropy loss for classification problems. This loss measures the discrepancy between the network's outputs and the actual values. To enhance accuracy, the network typically uses backpropagation of the error, which enables modification of weights and bias about the gradient of the loss function. This involves figuring out how the weights influence the loss and developing a way of adjusting them to minimize the error, which may employ gradient descent or any of those derived. Neural networks are learned in batches over several epochs, which helps optimize the model's parameters. The architecture of the neural networks is shown in Fig. 5. It is a moderately valuable tool in the contemporary development of AI and ML techniques to perform various data analyses.

A feed-forward neural network calculates outputs based on a set of inputs given to the layer in the network. All layers take input data, weight bias, and an activation function to generate an output. First, the raw data enters the network into the input layer. The weights then scale this data, sum it up with the corresponding biases, and introduce it to an activation function, which gives out the output layer and becomes the input to the next layer. It goes on progressively, step by step, and at each step, the obtained transformed data is at a higher level of abstraction concerning the input. The last one produces the network's output that can be designed for various purposes, *e.g.*, classification or regression. Since the

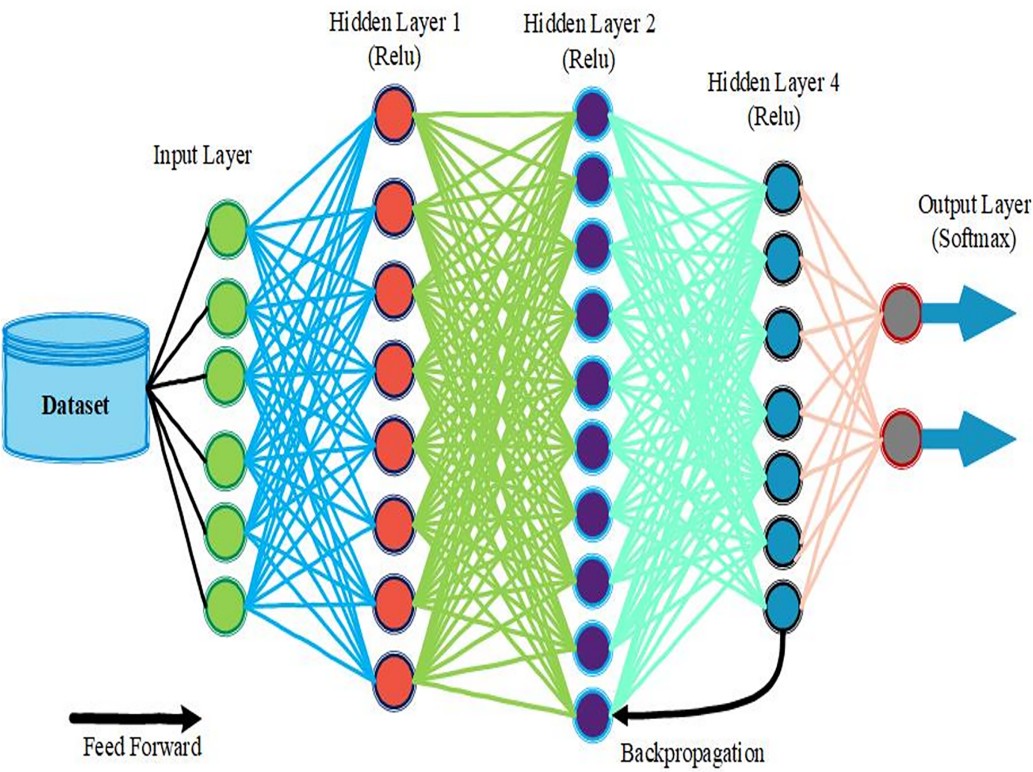

**Figure 5 Neural network structure and learning process.**

network operates on weights and biases, learning is achieved through errors determined through training data between the predicted and actual outputs. This series of transformations is capable of modeling the relationship between the values in the data set and, as such, makes neural networks a versatile tool in machine learning and AI. Neural networks include several principal mathematical operations and formulas that define their performance and learning approaches.

The machine learning algorithms for athlete performance monitoring and injury prediction require a focused analysis of specific models, hyperparameters, and training methods. CNNs excel in analyzing spatial data through multiple convolutional and pooling layers. Key hyperparameters, such as the number of filters, kernel size, and dropout rates, significantly impact model performance. Data augmentation techniques can also enhance training datasets, improving generalization. DNNs capture complex patterns within large datasets, comprising several hidden layers with varying neuron counts. Critical hyperparameters include the learning rate, batch size, and activation functions (*e.g.*, ReLU, Sigmoid), which require careful tuning. Regularization methods, such as L2 regularization and dropout, help prevent overfitting. Training methods for both CNNs and DNNs may employ early stopping to maintain model effectiveness and optimizers like Adam or RMSprop for improved convergence. Comparing these deep learning approaches with traditional algorithms, such as SVM and random forests, highlights their superior capability in capturing intricate patterns from time-series data. This comparative analysis

underscores the advantages of employing deep learning techniques in athlete performance monitoring and injury prediction, demonstrating their relevance.

The neuron's output in a neural network is obtained by performing a weighted sum on the neurons' inputs and a bias term. This calculation is done for all neurons j in layer l and is the dot product of the input and weights of each input neuron, followed by adding a bias. Algorithm 1 depicts Eq. (13), which is the workflow of a neural network.

$$z_j(l) = \sum_i w_{ij}(l)a_i(l-1) + b_j(l). \tag{13}$$

The biases and weights are updated with gradient descent to minimize the loss function. For a weight $w$, the update rule is Eq. (14):

$$w = w - \eta \frac{\partial loss}{\partial w}. \tag{14}$$

Backpropagation computes the gradient of the loss function to each weight by applying the chain rule. For each weight, the gradient is given by Eq. (15):

$$\frac{\partial loss}{\partial w_{ij}^{(l)}} = \delta_j^{(l)} a_i^{(l-1)} \tag{15}$$

where, $\delta_j^{(l)}$, is the error term for neuron $j$ in layer l, which is computed using:

$$\delta_j^{(l)} = \frac{\partial loss}{\partial z_j^{(l)}} f'\left(z_j^{(l)}\right). \tag{16}$$

A feed-forward neural network calculates its output *via* a series of transformations across its layers. Each layer transforms the input it accepts into an output using a set of weights, biases, and an activation function, as calculated in Eq. (17).

$$\begin{aligned} h^{(1)} &= f^{(1)}(W^{(1)}x + b^{(1)}) \\ h^{(l)} &= f^{(l)}(W^{(l)}h^{(l-1)} + b^{(l)}) \; for \; l = 2,\dots,L \\ \hat{y} &= f^{(L+1)}(W^{(L+1)}h^{(L)} + b^{(L+1)}) \end{aligned} \tag{17}$$

where $W^{(l)}$ and $b^{(l)}$, are the weights and biases of the $l^{th}$ layer, and $f^{(l)}$, is the activation function.

## Performance evaluation matrix

A performance evaluation matrix is a business performance matrix used to analyze and judge the results of a system Model or processes, thus presenting the performance in terms of effectiveness and efficiency. In disciplines like machine learning, project management, and business analytics, it assesses multiple performance indicators, including accuracy, precision, recall, F1 score, specificity, MAE, MSE, RMSE, t-tests and so on. One can classify the elements of the matrix into key indicators/scores/measures, target/benchmark values, and overall qualitative/rank/grade assessment. A few of these are decision-making since

---

**Algorithm 1 Neural network workflow.**

**Input:** {Training data $\{X_{train}, y_{train}\}$, Validation data $\{X_{val}, y_{val}\}$, Test data $\{X_{test}, y_{test}\}$, Model parameters and hyperparameters $\{\theta, \eta, B, E\}$}

**Output:** {Trained model M}

**Steps:**

1. **Data Preparation**

    $X_{train} \leftarrow \text{Normalize } (X_{train})$

    Divide into $\{X_{train}, y_{train}\}\{X_{val}, y_{val}\}\{X_{test}, y_{test}\}$

2. **Model Design**

    Layers $\{L_1, L_2, \ldots, L_n\}$ with activation functions $\{f_1, f_2, \ldots, f_n\}$

    Learning rate $\eta$, Batch size  B

3. **Initialize Weights:**

    $W_i \sim N(0, \sigma^2)$

4. **Training**

    **for** each Epoch e from 1 to E, **do**

      **for** each Batch b from 1 to B, **do**

        Compute activations $\mathbf{a}^{(l)}$ for each layer, l

        Calculate the loss function L (*e.g.*, Mean Squared Error or Cross-Entropy Loss)

        Compute gradients using the chain rule and update weights and biases:
        $\theta \leftarrow \theta - \eta \nabla_\theta L$

5. **Validation**

    Assess the model's performance on the validation set $\{X_{val}, y_{val}\}$

    Metrics: Accuracy, Precision, Recall, F1-Score, *etc.*

6. **Testing**

    Evaluate the final model on the test set $\{X_{test}, y_{test}\}$ to measure generalization
performance.

7. **Deployment**

    Integrate into a production environment.

8. **Monitoring**

    Track model performance and retrain if necessary.

      **end for**

    **end for**

---

relevant data from the market are collected, help the organization know its position compared to its competitors, and act as a tool for establishing various strengths and weaknesses that may be needed in the future. That being said, the creation and functioning of a performance evaluation matrix can be quite intricate, and issues such as criterion ambiguity and requirements for the availability of good-quality data must be considered to enhance the quality and validity of the assessment.

Accuracy measures the proportion of correct predictions out of all predictions made in Eq. (18):

$$Accu = \frac{TP + TN}{TP + TN + FP + FN}. \tag{18}$$

Precision measures the proportion of true positive predictions out of all positive predictions made in Eq. (19):

$$Precision = \frac{FN}{TP + FP}. \tag{19}$$

Recall (also known as sensitivity) measures the proportion of true positive predictions out of all actual positive instances given in Eq. (20):

$$Recall = \frac{TP}{TP + FN} \tag{20}$$

The F1 score is the harmonic mean of precision and recall as calculations Eq. (21):

$$F1\ Score = 2 \cdot \frac{Precision \cdot Recall}{Precision + Recall} \tag{21}$$

Specificity measures the proportion of true negative predictions out of all actual negative instances, as given in Eq. (22):

$$Specificity = \frac{TN}{TN + FN} \tag{22}$$

where:

- TP: True Positives
- TN: True Negatives
- FP: False Positives
- FN: False Negatives

MAE measures the average magnitude of errors in predictions without considering their direction, which is expressed in Eq. (23):

$$MAE = \frac{1}{n} \sum_{i=1}^{n} |y_i - \hat{y}_i|. \tag{23}$$

MSE measures the average of the squares of the errors are expressed in Eq. (24):

$$MSE = \frac{1}{n} \sum_{i=1}^{n} (y_i - \hat{y}_i)^2. \tag{24}$$

RMSE is the square root of the average of the squared differences between actual and predicted values expressed in Eq. (25):

$$RMSE = \sqrt{\frac{1}{n}\sum_{i=1}^{n}(y_1 - \hat{y}_i)^2}. \tag{25}$$

A t-test is used to determine if there is a significant difference between the means of the two groups. The formula for the t-test is expressed in Eq. (26):

$$t = \frac{\overline{X_1} - \overline{X_2}}{\sqrt{\dfrac{S_1^2}{n_1} + \dfrac{S_2^2}{n_2}}}. \tag{26}$$

A performance evaluation matrix is a vital tool to assess the effectiveness and efficiency of system models or processes across various fields, including machine learning and business analytics. It evaluates key performance indicators such as accuracy, precision, recall, F1 score, Specificity, MAE, MSE, RMSE, and t-tests. The matrix categorizes these metrics into key indicators, target values, and overall assessments, facilitating informed decision-making and competitive analysis. While it aids organizations in identifying strengths and weaknesses, its implementation can be complex, requiring attention to issues like criterion ambiguity and the need for high-quality data. Key performance metrics, including accuracy and precision, quantify predictive success, while MAE, MSE, and RMSE measure error magnitudes. Ultimately, the performance evaluation matrix is essential for enhancing organizational performance and guiding future improvements.

## EXPERIMENTAL RESULTS AND ANALYSIS

The results and analysis section presents a detailed analysis of the results likely to be obtained using the proposed model or system. This assessment is required to establish the model's applicability, effectiveness, and usefulness in an actual setting. It is intended to systematically evaluate the obtained result of all the experiments to diagnose whether the model developed can effectively solve the problem under consideration and how best it can be improved or if it is inferior in performance compared to any related remedy already proposed. The components of the experiment are establishing the test environments, data collection, and interpretation to conclude. The measures put in use for assessment are outlined because of the research goals and aims whenever necessary, for example, accuracy, preciseness, recall, and the like. Only the benefits and potential future work directions for the proposed approach are described, and the strengths and advantages of the presented approach are revealed, as well as the drawbacks and potential shortcomings. The experiment results begin with a detailed description of the experimental design, the results, and the analysis. It ensures that the outcomes become useful in solving the problems and provide a stable ground for further work in the field.

The application of the ISMS is based on neural networks, the configuration of the system architecture, and the requirement for system development and application, which

are vital conditions to measure the functionality and effectiveness of this intelligent system. A giant good processor like Intel Core i7 or even AMD Ryzen 7, which has four cores and eight threads, can at least mean a stomach for computing. The system should have 16 GB of DDR4 RAM installed, while for fast data input/output, the system should come with 512 GB SSD. To boost the graphical processing tasks, the laptop should contain an NVIDIA GTX 1660, as it is highly recommended. This software runs on Windows 10 and supports Python 3. 8 and neural network frameworks like TensorFlow2. 0, Keras, PyTorch, *etc*. Data processing and analysis tools applicable to sub-models are written using Python, and utilities employed include NumPy, Pandas, Scikit-Learn, SciPy, and Open Computer Vision. Jupyter Notebook is used as an integrated development environment for coding, while Hadoop/Spark handles big data management. Tableau, matplotlib, and Seaborn are tools that ideally aid in presenting the insights derived from data. The structure ensures in Table 1 that system requirements are adequate for developing, running, and analyzing the neural network models, establishing the solid ISMS base.

## Comparative analysis for anomaly detection

Table 2 displays time-series data collected from a wearable sensor system. It includes timestamps, heart rate (in beats per minute), body temperature (in degrees Celsius), acceleration (in g-forces), and GPS coordinates (latitude and longitude).

According to Table 2, at 10:00:00 on July 16, 2024, the recorded heart rate was 72 bpm, body temperature was 36.7 °C, acceleration was 0.5 g, and the GPS coordinates were 40.712776 (latitude) and −74.005974 (longitude). The data shows slight heart rate, body temperature, and acceleration variations over time, with consistent GPS coordinates indicating the data was collected in the exact location.

The environmental data captures various metrics and a distinct observation, each describing a different environmental variable.

Figure 6 is a dataset detailing environmental metrics with ten observations. It includes temperature (°C) ranging from −8 °C to 10 °C, showing the diversity in temperature conditions. Humidity (%) values span from −10% to 8%, with some negative values possibly indicating data processing or specific analytical contexts. The wind speed (km/h) ranges from −10 to 9 km/h, where negative values might reflect directional data or anomalies in measurement. Pressure (hPa) varies between −10 and 8 hPa, with negative numbers potentially used for specific analyses. Finally, rainfall (mm) ranges from −9 to 9 mm, where negative values might serve specific analytical purposes or data placeholders. This data aids in environmental analysis and understanding diverse weather conditions.

Physiological metrics are recorded during specific time intervals.

According to Fig. 7, from 10:00 to 10:05, the average heart rate was 73 bpm, with a body temperature of 36.8 °C and an average acceleration of 15.5 g, indicating moderate activity. In the next interval, from 10:05 to 10:10, the average heart rate rose slightly to 74 bpm, and body temperature increased to 36.9 °C, while average acceleration peaked at 16 g, suggesting heightened physical exertion. Between 10:10 and 10:15, the average heart rate decreased to 72 bpm, body temperature dropped slightly to 36.7 °C, and average acceleration reduced to 14.5 g, indicating a decrease in activity. Finally, from 10:15 to

**Table 1 System configuration and requirements.**

|  | Component | Specification | Details |
|---|---|---|---|
| Hardware | Processor | Intel Core i7 or AMD Ryzen 7 | Minimum four cores, eight threads |
|  | RAM | 16 GB DDR4 | Minimum requirement for smooth operation |
|  | Storage | 512 GB SSD | Fast read/write speeds for efficient data handling |
|  | Graphics card | NVIDIA GTX 1660 | For processing intensive graphical tasks |
| Software | Operating system | Windows 10 | Ensure compatibility with software and libraries |
|  | Environment | Python 3.8, TensorFlow, Keras, PyTorch | Required for developing and running neural networks |
|  | Python libraries | NumPy, Pandas, Scikit-Learn, SciPy, OpenCV | For data manipulation, machine learning, and analysis |
|  | IDE | Jupyter Notebook | For coding and script execution |
|  | Data management tools | Hadoop/Spark | For big data processing and management |
|  | Data visualization tools | Tableau, Matplotlib, Seaborn | For visualizing data |

**Table 2 Time-series data of physiological and location metrics.**

| Timestamp | Heart rate (bpm) | Body temperature (°C) | Acceleration (g) | GPS latitude | GPS longitude |
|---|---|---|---|---|---|
| 2024-07-16 10:00:00 | 72 | 36.7 | 0.5 | 40.712776 | −74.005974 |
| 2024-07-16 10:00:05 | 75 | 36.9 | 0.7 | 40.712776 | −74.005974 |
| 2024-07-16 10:00:10 | 78 | 37.1 | 0.6 | 40.712776 | −74.005974 |
| 2024-07-16 10:00:15 | 70 | 36.8 | 0.5 | 40.712776 | −74.005974 |
| 2024-07-16 10:00:20 | 73 | 36.9 | 0.6 | 40.712776 | −74.005974 |

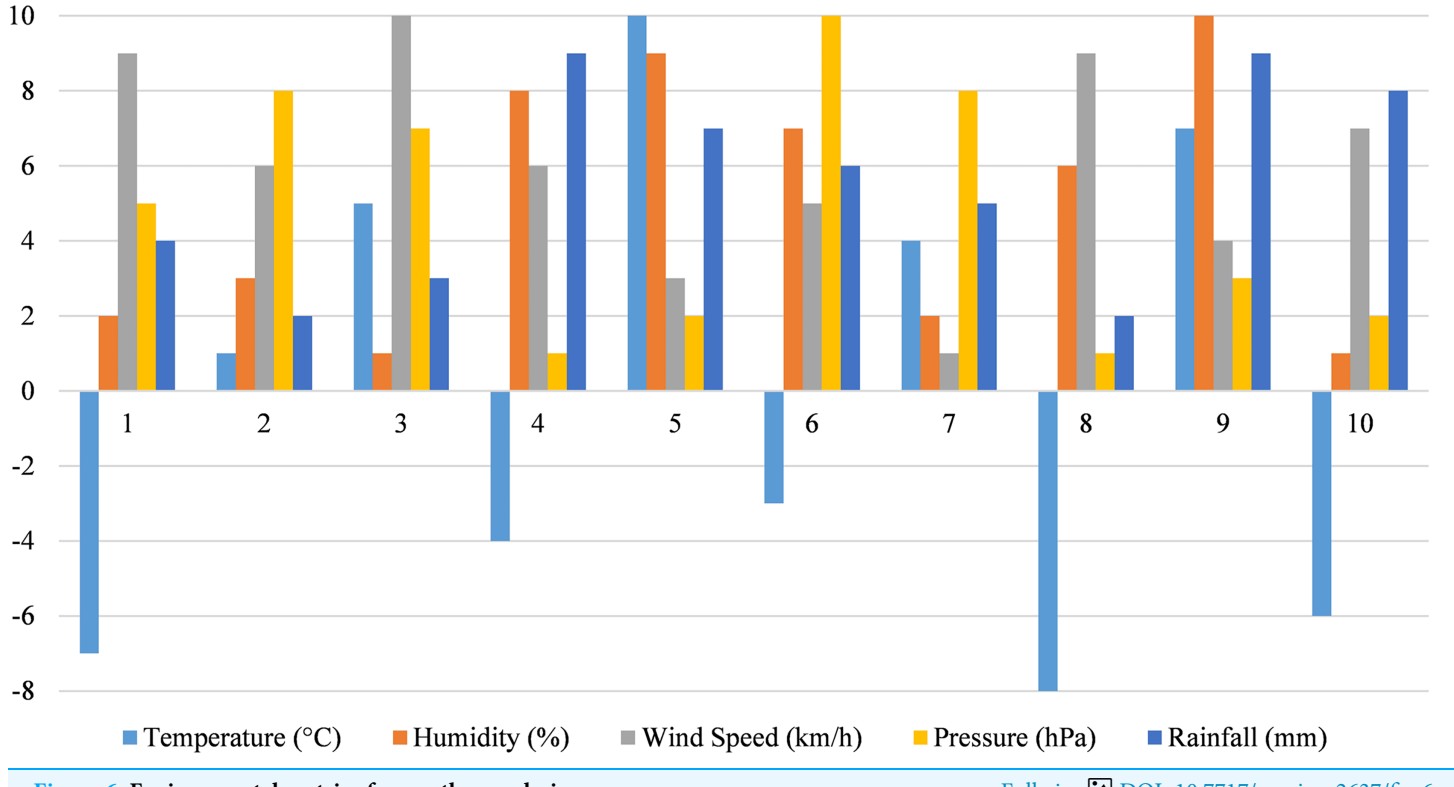

**Figure 6 Environmental metrics for weather analysis.**

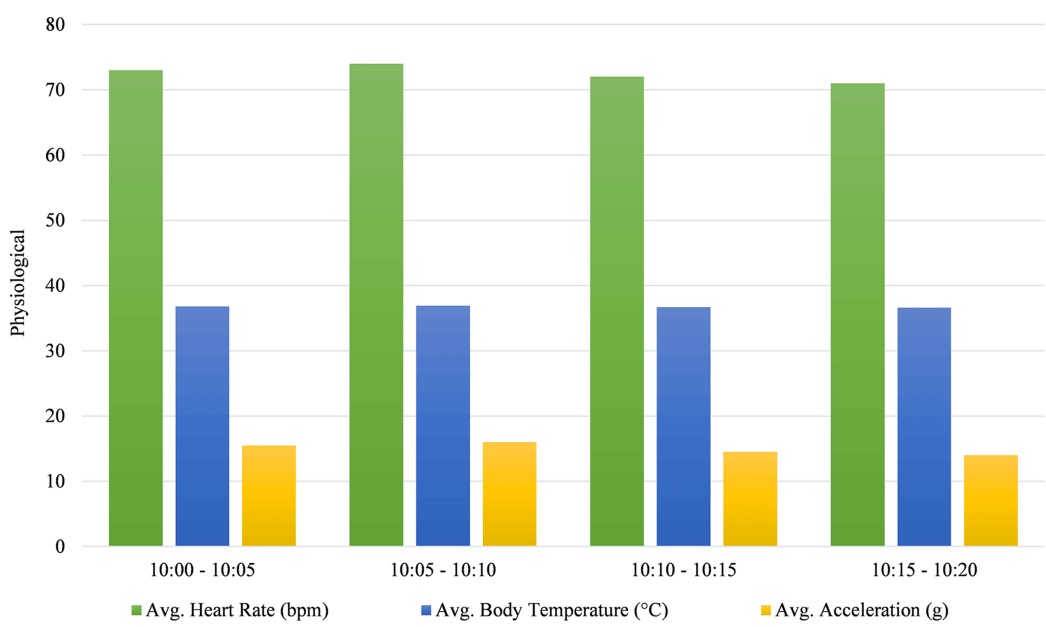

**Figure 7** **Average physiological metrics during specified time intervals.**

10:20, the heart rate declined to 71 bpm, body temperature fell to 36.6 °C, and acceleration decreased to 14 g, reflecting reduced physical activity.

## Comparative examination of machine learning models

The performance of predictive models is strongly influenced by the types of sensors used to gather data. Figure 8 below illustrates how various combinations of sensors affect key performance metrics, including accuracy, precision, recall, F1 score, and specificity.

As seen in Fig. 8, it is observed that as more types of sensors were incorporated, the performance of the developed model was enhanced. For instance, when taking heart rate and acceleration sensors together, the accuracy is 0.88, which rises when blood pressure sensors are added at 0.92. Regarding accuracy, the performance is the highest when all the sensors are incorporated, and the accuracy is at 0.94, precision is 0.95, recall is 0.94, F1 is 0.94, and specificity is 0.95. Using multiple sensors improves the models' capacity to estimate results by offering additional data points.

The performance of models based on various sensor configurations is shown in Fig. 9. The heart rate sensor generalizes moderate accuracy with MAE at 15 and RMSE at 0.2. The acceleration sensor can be identified as providing better results. Its MAE is 0.12, and RMSE is 0.17. The temperature sensor is less accurate, with an MAE of 0.16 and RMSE of 0.22, while the blood pressure sensor is the most accurate, with the lowest MAE of 0.08 and RMSE of 0.1. Integrating all the sensors used here provides the best performance, with the least MAE of 0.06 and the least RMSE of 0.09.

The improvements noted at longer intervals stress the advantages of increasing the intervals between data analyses to get higher accuracy and stability of the models. Again, if the time interval has been increased from 5 to 30 s, the gain factors are quite distinct in all

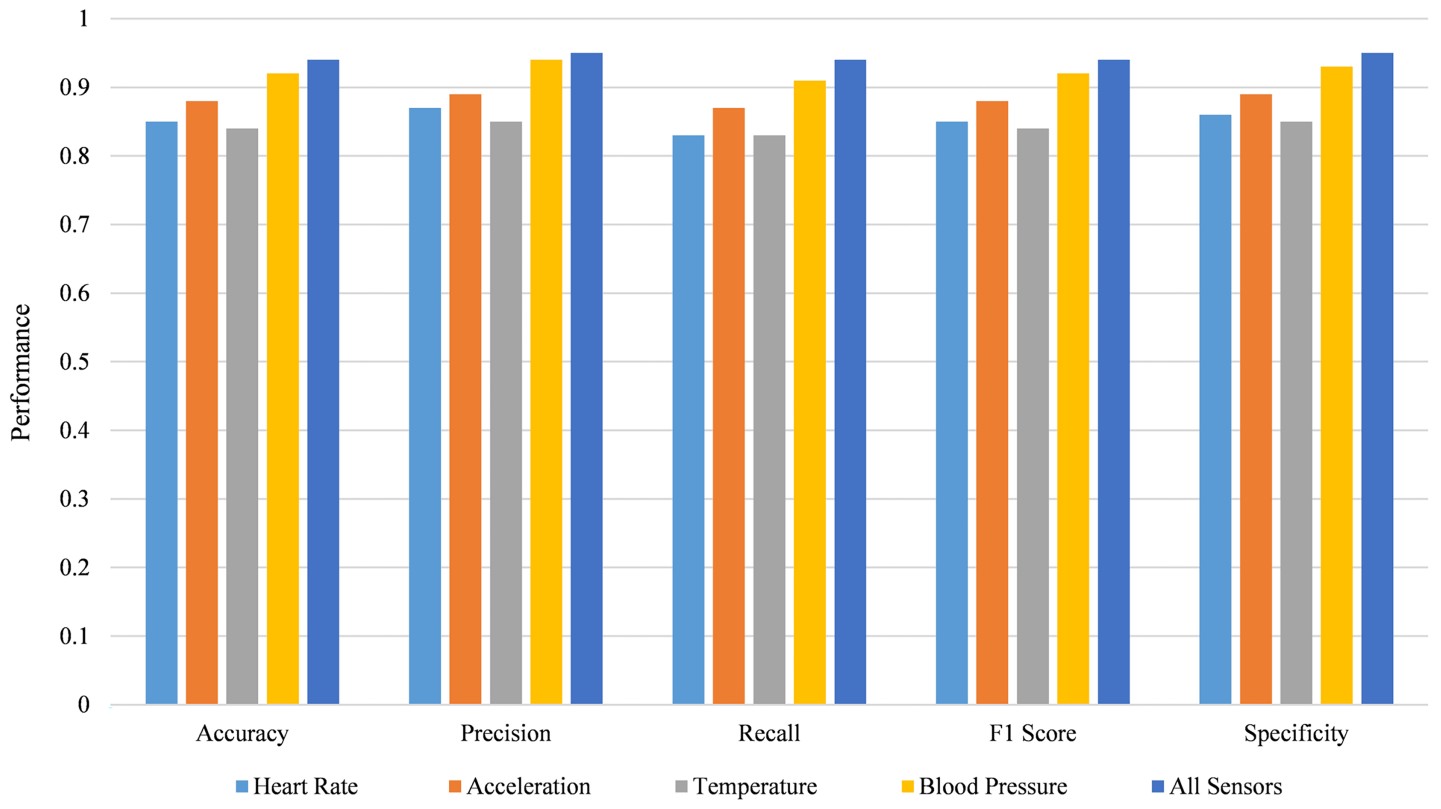

**Figure 8 Performance metrics of predictive models with various sensor combinations.**

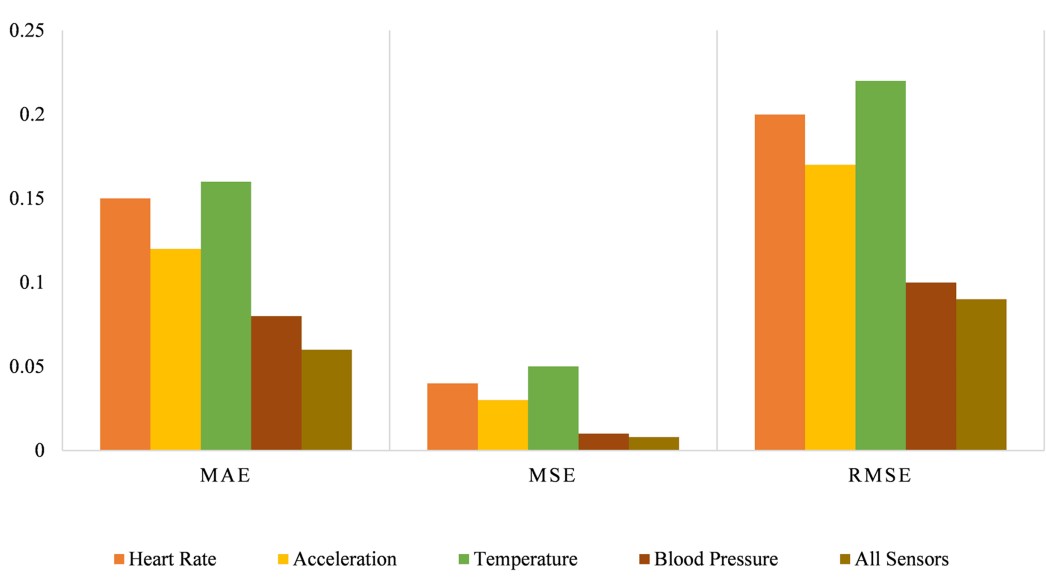

**Figure 9 Performance error metrics for different sensor hybrids.**
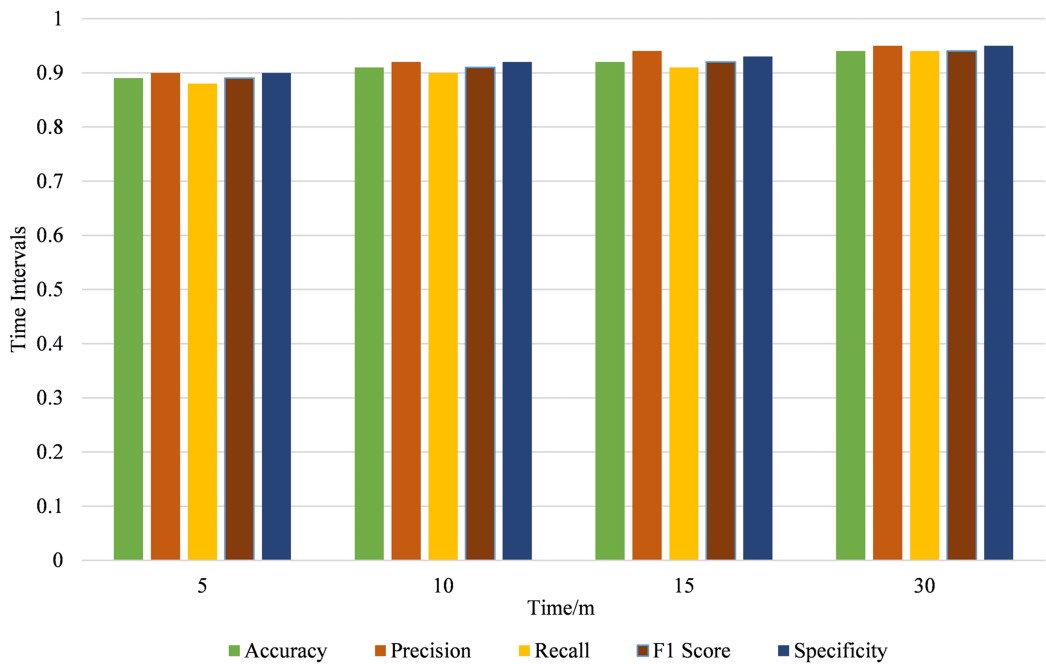

**Figure 10  Model performance metrics across different time intervals.**

the performance criteria. In particular, the model's effectiveness with specific metrics gradually increases accuracy, precision, recall, F1 score, and specificity. At 5 s, it has the best result, but with an increased interval of 30 s, the parameters are the highest. This indicates that longer intervals allow for more comprehensive data collection, leading to more reliable and accurate model predictions. Figure 10 below illustrates model performance metrics across various time intervals.

The performance metrics for different time intervals reveal the following trends: As the time interval increases, the MAE decreases from 0.11 to 0.06, indicating improved accuracy. Similarly, the MSE and RMSE decrease, reflecting reduced prediction error. The t-test values also increase, suggesting greater statistical significance. Figure 11 shows the longer time intervals lead to more accurate and reliable performance metrics.

In the suggested model, while performance remains robust with increasing sensor nodes, there is a gradual decline in significance as the data load grows. With 10 nodes, the system achieves increased performance across all metrics, including accuracy (0.98), precision (0.96), recall (0.94), F1 score (0.97), and specificity (0.99). As the number of nodes increases to 50, performance slightly decreases but remains high, with accuracy at 0.97. With 100 nodes, the performance shows a further slight decline, and with 500 nodes, the metrics stabilize at a lower level, with an accuracy of 0.95. Figure 12 illustrates the performance metrics of a system as the number of sensor nodes varies.

This movement shows that as data load increases, error metrics generally worsen, but the statistical significance measured by the t-test becomes less prominent. For a data load of 10 sensor nodes, the MAE is 0.02, MSE is 0.004, RMSE is 0.06, and t-test value is 0.55. As the number of sensor nodes increases to 50, MAE rises to 0.03, MSE to 0.007, RMSE to

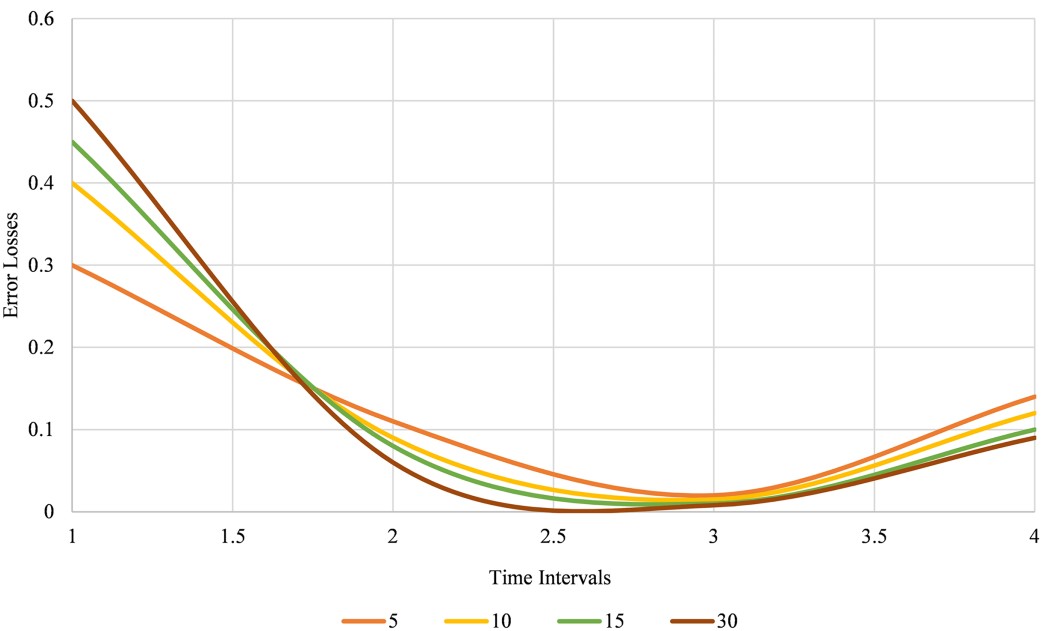

**Figure 11 Performance error metrics across different time intervals.**

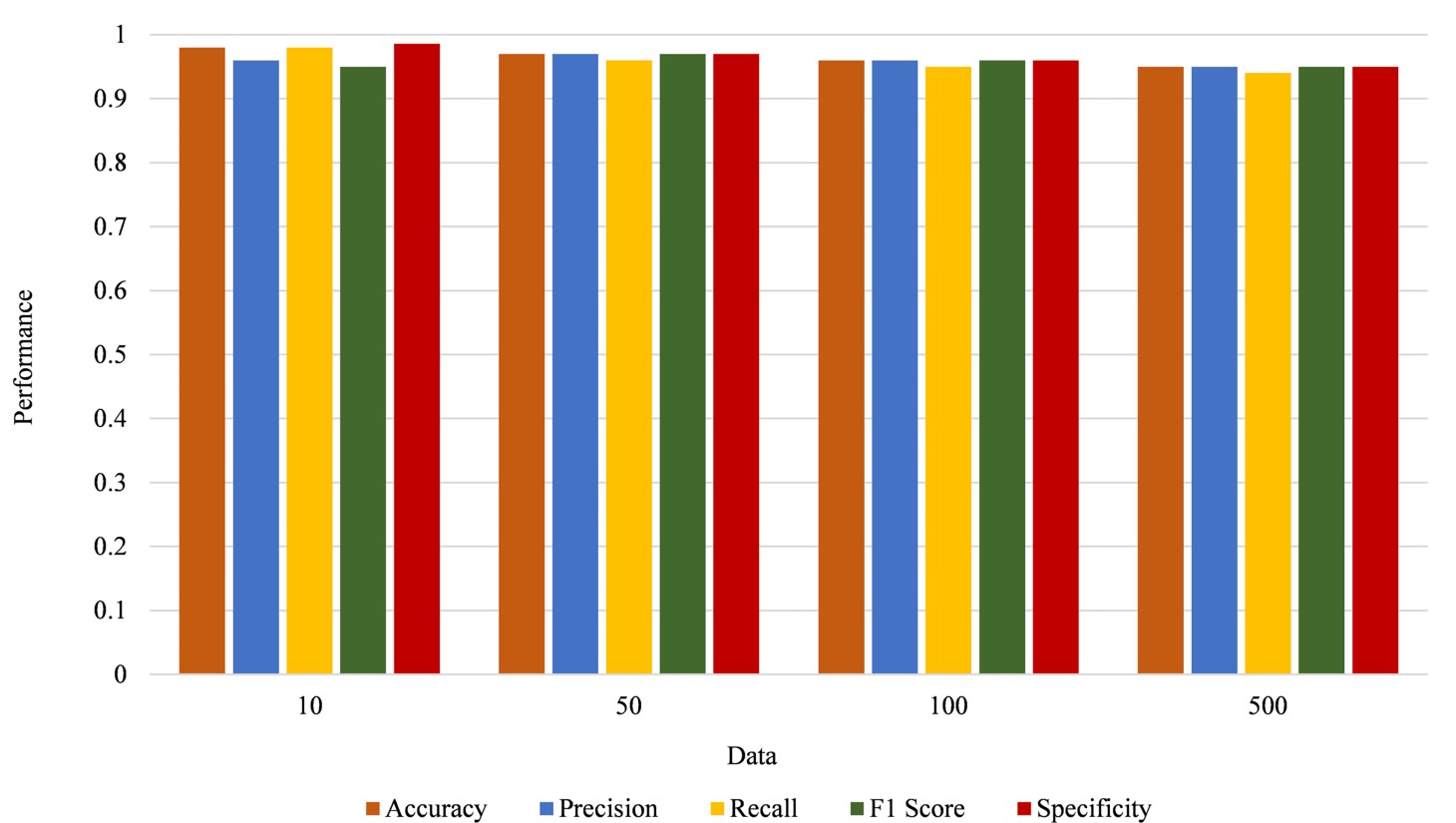

**Figure 12 Performance metrics of the system with varying data load.**

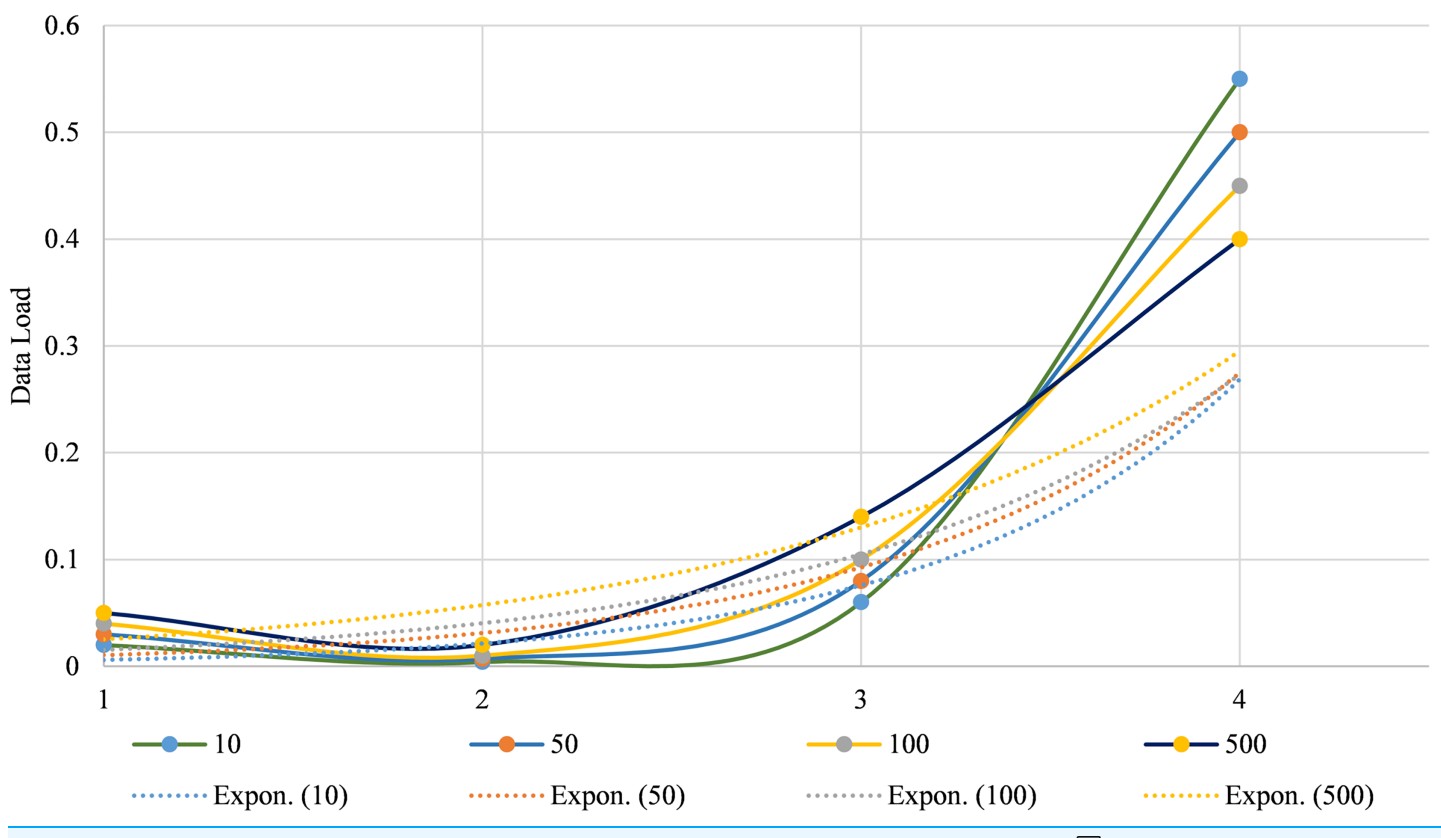

**Figure 13 Performance error metrics with increasing data load.**

0.08, and t-test decreases to 0.5. At 100 nodes, MAE reaches 0.04, MSE 0.01, RMSE 0.1, and t-test is 0.45. For the most significant data load of 500 nodes, MAE increases to 0.05, MSE to 0.02, RMSE to 0.14, and the t-test value decreases to 0.4. Figure 13 shows the performance error metric, MAE, MSE, Root RMSE, and t-test values, which vary with different data loads.

The comparative analysis of different anomaly detection models provides various techniques for classification accuracy and error metrics. The focus is evaluating several models based on multiple performance indicators, including accuracy, precision, recall, F1 score, specificity, and various error metrics such as MAE, MSE, and RMSE. This analysis helps to identify which model offers the best balance between detection performance and error rates, guiding the choice of model based on specific application needs.

According to Table 3, the analysis indicates that neural networks lead in performance with an accuracy of 0.94, precision of 0.93, recall of 0.91, and an F1 score of 0.95, which signifies the highest effectiveness in detecting anomalies. Additionally, they exhibit the lowest MAE (0.06), MSE (0.008), and RMSE (0.09), demonstrating minimal error in predictions. Autoencoders follow closely, achieving high accuracy (0.93) and precision (0.94), with slightly higher MAE (0.07) and MSE (0.01). One-class SVM and local outlier

**Table 3 Comparative analysis of anomaly detection models for different performance metrics.**

| Model | Accuracy | Precision | Recall | F1 score | Specificity | MAE | MSE | RMSE |
|---|---|---|---|---|---|---|---|---|
| Neural networks | 0.94 | 0.93 | 0.91 | 0.95 | 0.97 | 0.06 | 0.008 | 0.09 |
| Autoencoders | 0.93 | 0.94 | 0.93 | 0.93 | 0.94 | 0.07 | 0.01 | 0.10 |
| One-class SVM | 0.88 | 0.89 | 0.88 | 0.88 | 0.89 | 0.12 | 0.03 | 0.17 |
| Local outlier factor | 0.90 | 0.91 | 0.90 | 0.90 | 0.91 | 0.10 | 0.02 | 0.14 |

factor show lower metrics, with one-class SVM having the highest MAE (0.12) and MSE (0.03) and local outlier factor presenting a balance between performance and error rates.

Figure 14 compares various optimization algorithms, focusing on their performance metrics: accuracy, precision, recall, F1 score, and specificity. Adam displays the most remarkable performance with an accuracy record of 0.94 and a high precision of 0.93, recall of 0.91, F1 score of 0.95, and specificity of 0.97. SGD trails with a comparatively reliable accuracy of 0.89, a good precision of 0.90, and a recall of 0.88, although less than Adam. Like SGD, RMSprop achieves the test set's accuracy of 0.91, precision of 0.92, and recall of 0.91. On the other hand, the scores for the two options, namely Adagrad and Adadelta, are lower; Adagrad attained an accuracy of 0.87 and Adadelta 0.88. Finally, FTRL, Nadam, Rprop, and L-BFGS have comparatively low performance based on the presented accuracy rates of 0.72 to 0.77, which shows these algorithms do not work as efficiently as the others in attaining the given assignments.

The statistical significance of modifications, with lower values, generally signifies more reliable results. Adam demonstrates the lowest MAE of 0.06, indicating the best prediction accuracy, and also has the lowest mean MSE of 0.008 and RMSE of 0.09, reflecting minimal prediction error. In contrast, L-BFGS shows the highest MAE of 0.22, MSE of 0.07, and RMSE of 0.26, suggesting less effective performance. Figure 15 compares optimization algorithms using performance metrics.

The proposed model, represented by the confusion matrix, indicates its classification significance across classes. Class 0 reached 282 correct classifications, with 18 misclassified as Class 1, showing good performance but some confusion between Class 0 and Class 1. For Class 1, it accurately predicted 264 samples, with 36 misclassified as Class 2, indicating some overlap with Class 2 but no misclassifications as Class 0. For Class 2, it correctly identified 358 samples, with 24 misclassified as Class 0 and 18 as Class 1, reflecting some confusion with the other two classes. The model shows substantial accuracy and precision, particularly excelling in predicting Class 2 and demonstrating a high accuracy of 90.4%. The main area for improvement is reducing misclassifications between Class 0 and Class 1, as shown in Fig. 16.

This comparison highlights the strengths and weaknesses of each model in various aspects of performance. Random forest follows closely with robust scores, showing its reliability but slightly lower than neural networks in precision and specificity. Support vector machine and K-nearest neighbors also perform well but exhibit slightly lower metrics than random forest. Logistic regression, while effective, shows the lowest
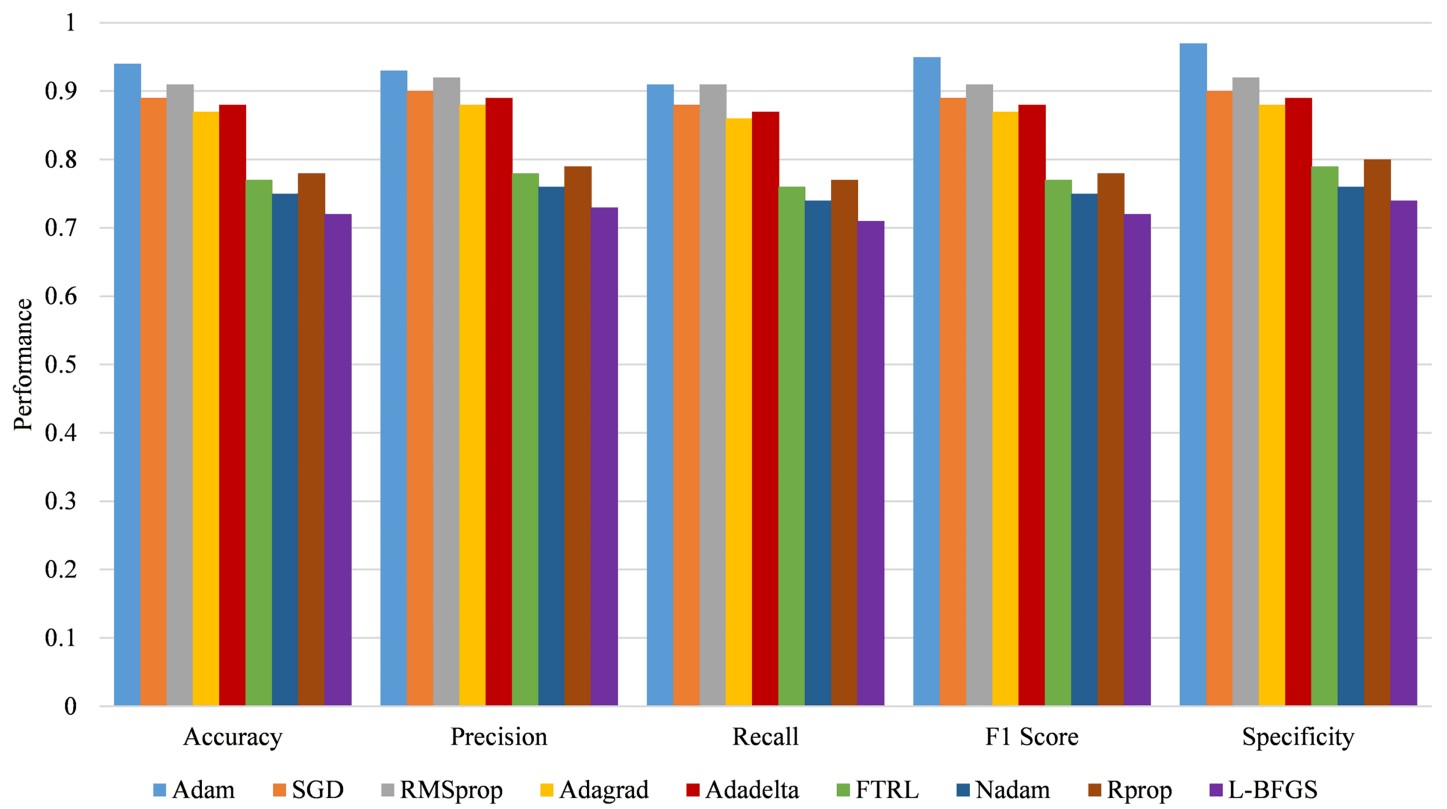

**Figure 14** Performance comparison of optimization algorithms.     

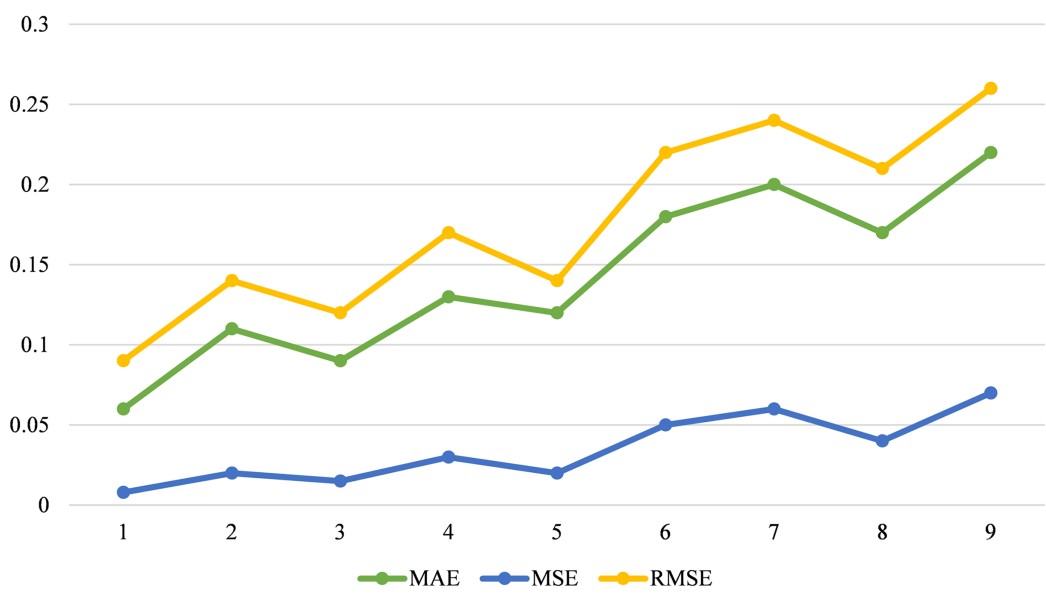

**Figure 15** Comparison of optimization algorithms using error metrics.

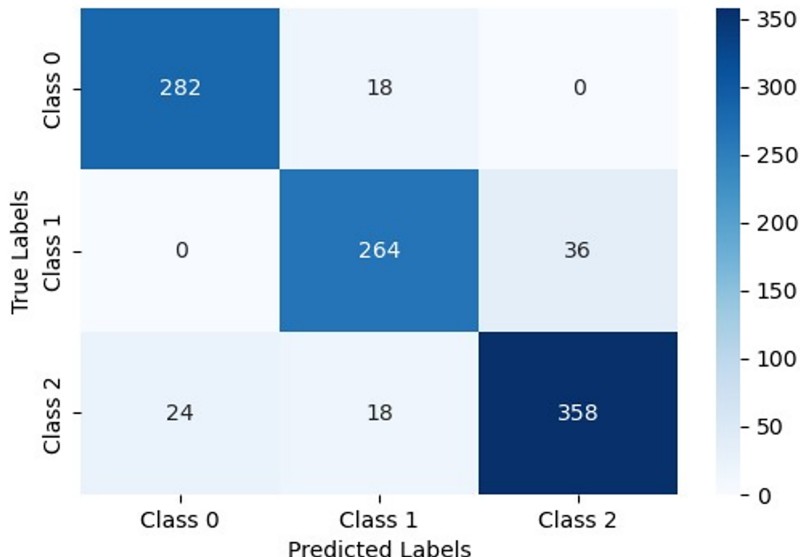

**Figure 16 Confusion matrix for the proposed model performance.**

performance among the models, particularly in accuracy and F1 score. The Neural Network model demonstrates the highest accuracy (0.94), precision (0.93), recall (0.91), F1 score (0.95), and specificity (0.97), indicating superior performance in detecting and classifying data correctly. Figure 17 compares different models' performance, evaluating their effectiveness across various metrics.

The receiver operating characteristic curves for the proposed models, including random forest, support vector machine, and K-nearest neighbors. The ROC curves demonstrate the performance of each model across various threshold levels. The neural network model outperforms the others, achieving the highest area under the curve (AUC). Figure 18 indicates that the neural network model has superior classification ability, offering better true positive rates than the false positive rates compared to random forest, SVM, and KNN.

The statistical analysis shows the significance of differences in performance among these models. Figure 19 illustrates the performance of various models using error metrics. The neural network model exhibits the lowest MAE, MSE, and RMSE, indicating superior accuracy and error handling compared to other models. The random forest model follows with slightly higher errors. Support vector machine and K-nearest neighbors show moderate performance, with logistic regression having the highest errors among these models.

To compare and evaluate different performance metrics models in sports and health monitoring technologies, optimize the accuracy and significance of predictive models employed for athlete performance analysis, and compare various approaches to identify which models offer the best balance of accuracy, precision, recall, F1 score, and specificity. Table 4 presents a comparative analysis of these metrics across several studies, including the proposed scheme.
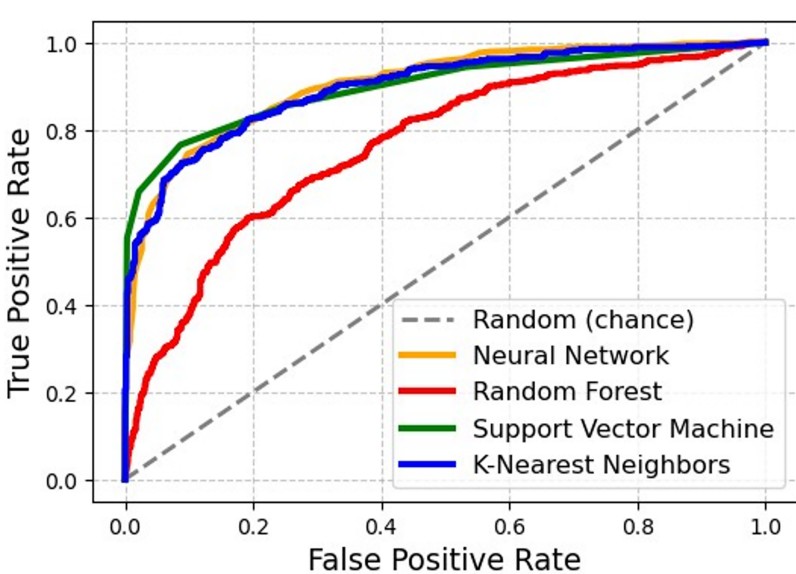

**Figure 17 Comparative performance analysis of various machine learning models.**

**Figure 18 ROC curve comparison for classification models.**

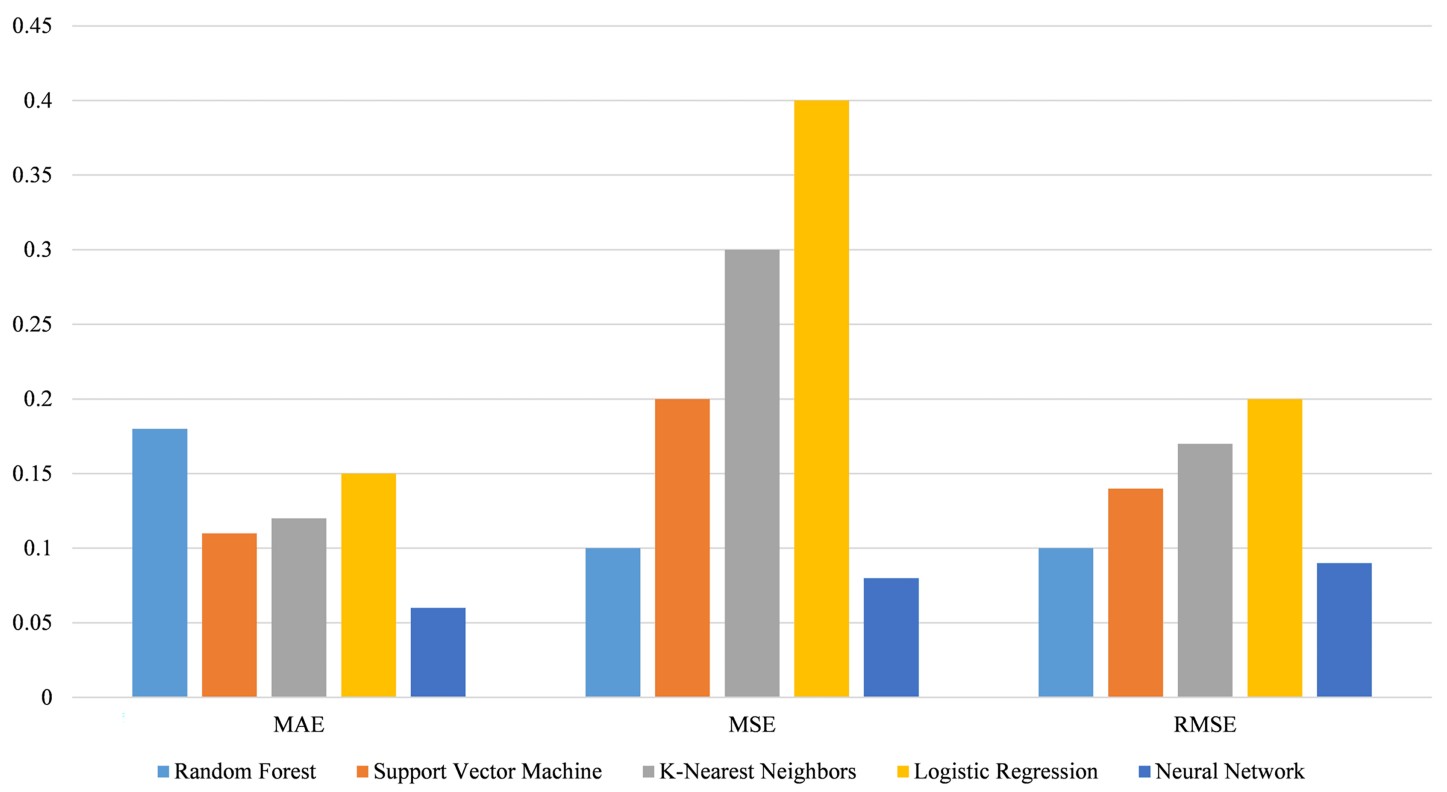

**Figure 19 Performance comparison of models using error metrics and statistical significance.**

**Table 4 Comparative analysis of performance metrics for predictive models in sports monitoring.**

| Study | Accuracy | Precision | Recall | F1 score | Specificity |
|---|---|---|---|---|---|
| Proposed scheme | 0.94 | 0.93 | 0.91 | 0.95 | 0.97 |
| *Yang & Tang (2022)* | 0.84 | 0.82 | 0.80 | 0.81 | 0.85 |
| *Dang et al. (2023)* | 0.83 | 0.80 | 0.78 | 0.79 | 0.82 |
| *Zhang et al. (2023)* | 0.85 | 0.83 | 0.81 | 0.82 | 0.86 |
| *Li et al. (2022)* | 0.82 | 0.79 | 0.76 | 0.77 | 0.83 |

According to Table 4, the comparison highlights the superior performance of the proposed scheme relative to other studies. It achieves an accuracy of (0.94), precision of (0.93), recall of (0.91), F1 score of (0.95), and specificity of (0.97), significantly surpassing the metrics of other models. For instance, *Yang & Tang (2022)* reported an accuracy of (0.84) and an F1 score of (0.81), while *Zhang et al. (2023)* achieved a slightly lower accuracy of (0.85) compared to the proposed scheme significance in providing higher predictive accuracy and reliability in performance metrics.

## CONCLUSION

This article has proposed a novel intelligent sports management system using WSNs with state-of-the-art machine learning algorithms to improve athlete performance and health monitoring. The suggested approach addressed basic regulations in traditional sports monitoring techniques by including a thorough framework for real-time analysis and predictive abilities. At the core of the chosen method are high-efficiency, low-powered aloft sensors strategically established to monitor performance metrics. These sensors feed data to a central processing hub, which, combined with cloud-based storage, enables beneficial data investment, protection, and analysis. The system effectively mitigated energy consumption and data integrity issues in dynamic sports environments and used low-energy wireless protocols. The proposed study has applied machine learning algorithms such as random forests, support vector machines, and neural networks to identify performance patterns and detect anomalies. These abilities facilitate real-time data visualization, injury alerts, and trend analysis, thereby supporting athletes, coaches, and medical staff in optimizing performance and preventing injuries. Using exclusive databases and cloud storage improves the method's efficiency and scalability in addressing large datasets. Improving the efficiency and accuracy of machine learning algorithms through advanced techniques, such as deep learning and ensemble methods, is crucial for further improving predictive accuracy and anomaly detection. Future work should focus on these areas to refine the system and maximize its impact on sports management and athlete health. To strengthen the ISMS model's practical relevance, future work will focus on validating it with real-world athlete data, particularly from professional and semi-professional sports. This will provide insight into the model's performance beyond simulations. Additionally, the model will be enhanced by incorporating advanced machine-learning techniques and developing a real-time data pipeline for live athlete tracking. Expanding the dataset across various sports will further improve the model's generalizability, refining it as a robust sports performance and injury prevention tool.

### Funding

The author received no funding for this work.

### Competing Interests

The author has no competing interests.

### Author Contributions

- ZhiGuo Zhu conceived and designed the experiments, performed the experiments, analyzed the data, performed the computation work, prepared figures and/or tables, authored or reviewed drafts of the article, and approved the final draft.

### Data Availability

The dataset and Python code used for implementation are available in the Supplemental File.

The third-party dataset is available at Mendeley: Burns, Alon; Wallot, Sebastian; Berson, Yair; Gordon, Ilanit (2022), "Dataset of physiological, behavioral, and self-report measures from a group decision-making lab study", Mendeley Data, V4, doi: 10.17632/w8h752hbw5.4.

## Supplemental Information

Supplemental information for this article can be found online at http://dx.doi.org/10.7717/peerj-cs.2637#supplemental-information.

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
