# Peer review of "Design and implementation of an intelligent sports management system (ISMS) using wireless sensor networks"

_PeerJ Computer Science, doi:10.7717/peerj-cs.2637_

## Round 0.1 · original submission · Major Revisions

· Academic Editor

Major Revisions

Both reviewers recognize the manuscript's merit, yet they have identified several issues that must be addressed in the revised version.

One significant concern is the need for a clearer and more precise definition of research problem. The current version of the manuscript does not sufficiently present the specific challenges being addressed, which makes it difficult for readers to fully understand the scope and importance of the research.

In addition, the reviewers pointed out that certain sections of the manuscript lack sufficient detail, e.g., the description of the Data Management Layer. This section needs to be expanded with more comprehensive explanations to provide a clearer understanding of its structure, components, and functionality, ensuring readers can understand its role within the overall system.

Another concern pertains to the generality of the proposed approach. One of the reviewers questioned whether the approach is applicable across different sports or if it is specifically tailored to a particular type of sport. This issue must be addressed or justify its focus on a specific domain.

Lastly, one reviewer suggested that additional experiments are needed to thoroughly analyze how various optimizations influence the system's performance. This would involve more testing to evaluate the impact of different optimization techniques on key performance metrics, thereby strengthening the manuscript's empirical contribution.

Addressing these points in the revised version will significantly improve the clarity, depth, and generalizability of the research.

Reviewer 1 ·

Basic reporting

No Comments

Experimental design

No Comments

Validity of the findings

No Comments

Additional comments

Review comments to enhance the quality of the research work titled "Design and Implementation of an Intelligent Sports Management System (ISMS) Using Wireless Sensor Networks":

1. The description of the Data Management Layer is not sufficiently detailed. It would be beneficial to explain how the system ensures data integrity, manages data storage, and handles large volumes of real-time data from multiple athletes. Consider including data pipeline architecture and a more comprehensive description of the database management techniques used.

2. The explanation of the machine learning algorithms (CNNs, DNNs) is too broad. A deeper discussion of the specific models, hyperparameters, and training methods used for athlete performance monitoring and injury prediction is necessary. It would also help to compare these models to other approaches in the literature.

3.While the paper demonstrates the effectiveness of the ISMS model through simulation, it lacks validation with real-world data. To enhance the credibility of the system, future experiments should include testing the model on real athlete data, perhaps from professional or semi-professional sports teams.

4. The choice of performance metrics such as MAE, MSE, and RMSE is appropriate for regression tasks, but it would be useful to explain why these specific metrics are used. Additionally, explain how these metrics apply to specific tasks in sports performance and injury prevention.

5. The security layer of the ISMS is briefly mentioned but lacks detail. Given the importance of protecting sensitive athlete data, a more comprehensive discussion of the encryption standards, data anonymization techniques, and authentication protocols used to safeguard data would significantly improve the security aspects of the system.

6. The paper does not adequately discuss how the ISMS scales when deployed across multiple venues, athletes, or sports organizations. It would be beneficial to explore how the system handles scalability challenges, including network congestion, increased data loads, and system stability under high demand.

7. Since the system relies on wearable sensors, the energy consumption and battery life of these devices are critical factors. Including an analysis of the power management strategies and energy consumption patterns of the wearable sensors would be useful. Highlighting how the system minimizes energy consumption in a sports setting would further enhance the system's practicality.

8. The ISMS is presented as a general system for sports management, but there is little discussion of its applicability to different sports. Consider including case studies or examples demonstrating how the system adapts to varying types of sports, such as team-based versus individual sports, contact versus non-contact sports, etc.

9. The paper does not provide a comparison between ISMS and other existing sports management or monitoring systems. Providing a comparative analysis of similar systems in the literature, focusing on advantages and shortcomings, would help to highlight the uniqueness and benefits of the proposed model.

10. The future work section should be expanded to discuss more concrete improvements, such as refining the neural network models, exploring new types of sensors (e.g., motion capture systems, EMG sensors), or integrating predictive analytics for long-term athlete health. Additionally, the paper could benefit from a discussion of potential system extensions, such as incorporating virtual reality (VR) for immersive athlete training simulations.

11. While the paper emphasizes real-time monitoring, there is little detail on how real-time data processing is achieved. Including details on how the system processes data in real time, whether through edge computing, cloud computing, or a combination of both, would enhance the technical rigor of the work.

12. The optimization strategies used to ensure that the ISMS performs efficiently on different hardware platforms (e.g., Intel i7, AMD Ryzen) are mentioned, but the paper would benefit from further exploration of how these optimizations affect the system's overall performance, especially in mobile and embedded environments.

By addressing these technical review comments, the research paper can be significantly improved in terms of detail, depth, and relevance to the broader research community.

·

Basic reporting

The article describes a system for the design and implementation of an intelligent sports
management system (ISMS) using wireless sensor networks (WSN)
The wireless networks infrastructure is very well described and analyzed.
The sensors indicates mesures very basic parameters, like hearth rate, temeperature, acceleration, and it is very unlikely to obtain information or indications on the performance of an athlete using these lonely parameters, also using neural networks or other AI instruments.
Then, the goal of the study is quite confused, including both performance improvements and safety/health prevention improvements.
In both cases a fiurther number of parameters are to be measured and inferenced to gain a prevention program or an optimization of training for better athletic performances.
FAILS

Experimental design

Is not presented an experimental strategy and planning, including metrics, expected results and mitigations for failure risks.
FAILS

Validity of the findings

Findings and descriptions in the WNS issues are very appreciable. Findings and goals in the ISMS issues are poor and not clear, especially about phisiological parameters to be monitored and analyzed by an AI algorithm.
FAILS

---

## Round 0.2 · accepted · Accept

· Academic Editor

Accept

The authors have responded to the reviewers' comments thoroughly and satisfactorily, addressing the concerns raised during the review process. Their revisions have significantly enhanced the quality and clarity of the manuscript. As a result, the paper is now suitable for acceptance and publication in the PeerJ Computer Science journal.

Reviewer 1 ·

Basic reporting

All the queries were addressed.

Experimental design

no comment

Validity of the findings

no comment

Additional comments

no comment